# ROI Maximization
# in Stochastic Online Decision-Making

**Nicolò Cesa-Bianchi**
Università degli Studi di Milano & DSRC
nicolo.cesa-bianchi@unimi.it

**Tommaso Cesari**
Toulouse School of Economics
tommaso.cesari@tse-fr.eu

**Yishay Mansour**
Tel Aviv University & Google research
mansour@tau.ac.il

**Vianney Perchet**
CREST, ENSAE & Criteo AI Lab
vianney.perchet@normalesup.org

## Abstract

We introduce a novel theoretical framework for Return On Investment (ROI) maximization in repeated decision-making. Our setting is motivated by the use case of companies that regularly receive proposals for technological innovations and want to quickly decide whether they are worth implementing. We design an algorithm for learning ROI-maximizing decision-making policies over a sequence of innovation proposals. Our algorithm provably converges to an optimal policy in class $\Pi$ at a rate of order $\min\left\{1/(N\Delta^2), N^{-1/3}\right\}$, where $N$ is the number of innovations and $\Delta$ is the suboptimality gap in $\Pi$. A significant hurdle of our formulation, which sets it aside from other online learning problems such as bandits, is that running a policy does not provide an unbiased estimate of its performance.

## 1   Introduction

Often, companies have to make yes/no decisions, such as whether to adopt a new technology or retire an old product. However, finding out the best option in all circumstances could mean spending too much time or money in the evaluation process. If the decisions to make are many, one could be better off making more of them quickly and inexpensively, provided that these decisions have an overall positive effect. In this paper, we investigate the problem of determining a decision policy to balance the reward over cost ratio optimally (i.e., to maximize the return on investment).

**A motivating example.**   Consider a technology company that keeps testing innovations to increase some chosen metric (e.g., benefits, gross revenue, revenue excluding the traffic acquisition cost). Before deploying an innovation, the company wants to figure out whether it is profitable. As long as each innovation can be tested on i.i.d. samples of users, the company can perform randomized tests and make statistically sound decisions. However, there is an incentive to make these tests run as quickly as possible because, for example, the testing process is expensive. Another reason could be that keeping a team on a project that has negative, neutral, or even borderline positive potential prevents it from testing other ideas that might lead to a significantly better improvement. In other words, it is crucial to learn when to drop barely positive innovations in favor of highly positive ones, so to increase the overall flow of improvement over time (i.e., the ROI of the tests).

More generally, our framework describes problems where an agent faces a sequence of decision tasks consisting of either accepting or rejecting an innovation. Before making each decision, the agent can invest resources into reducing the uncertainty on the value brought by the innovation. The global objective is to maximize the total ROI. Namely, the ratio between the total value accumulated by accepting innovations and the total cost. For an in-depth discussion on alternative goals, we refer the reader to the Supplementary Material (Section D).

35th Conference on Neural Information Processing Systems (NeurIPS 2021).

**The model.** Each task $n$ in the sequence is associated with a pair $(\mu_n, D_n)$ that the learner can *never* directly observe.

- $\mu_n$ is a random variable representing the (possibly negative) true value of the $n$-th innovation.
- $D_n$ is a probability distribution over the real numbers with expectation $\mu_n$, modeling the feedback on the $n$-th innovation that the learner can gather from testing (see below).

During the $n$-th task, the learner can draw arbitrarily many i.i.d. samples $X_{n,1}, X_{n,2}, \ldots$ from $D_n$, accumulating information on the unknown value $\mu_n$ of the innovation currently being tested. After stopping drawing samples, the learner can decide to either accept the innovation, earning $\mu_n$ as a reward, or reject it and gain nothing instead. We measure the agent performance during $N$ tasks as the (expected) total amount of value accumulated by accepting innovations $\mu_n$ divided by the (expected) total number of samples requested throughout all tasks. In formulas, the ROI of a strategy is then

$$\frac{\sum_{n=1}^{N} \mathbb{E}\big[\mu_n \cdot \mathbb{I}\{\mu_n \text{ is accepted}\}\big]}{\sum_{n=1}^{N} \mathbb{E}\big[\text{number of samples requested during the } n\text{-th task}\big]}$$

In Section 3 we present this setting in more detail and introduce the relevant notation.

**I.I.D. assumption.** We assume that the value $\mu_n$ of the $n$-th innovation is drawn i.i.d. from an unknown and fixed distribution. This assumption is meaningful if past decisions do not influence future innovations whose global quality remains stable over time. In particular, it applies whenever innovations can progress in many orthogonal directions, each yielding a similar added value (e.g., when different teams within the same company test improvements relative to individual aspects of the company). If both the state of the agent and that of the environment evolve, but the ratio of good versus bad innovations remains essentially the same, then this i.i.d. assumption is still justified. In other words, it is not necessarily the absolute quality of the innovations that has to remain stationary, but rather the relative added value of the innovations given the current state of the system. This case is frequent in practice, especially when a system is close to its technological limit. Last but not least, algorithms designed under stochastic assumptions often perform surprisingly well in practice, even if i.i.d. assumptions are not fully satisfied or simply hard to check.

**A baseline strategy and policy classes.** A natural, yet suboptimal, approach for deciding if an innovation is worth accepting is to gather samples sequentially, stopping as soon as the absolute value of their running average surpasses a threshold, and then accepting the innovation if and only if the average is positive. The major drawback of this approach is that the value $\mu_n$ of an innovation $n$ could be arbitrarily close to zero. In this case, the number of samples needed to reliably determine its sign (which is of order $1/\mu_n^2$) would become prohibitively large. This would result is a massive time investment for an innovation whose return is negligible at best. In hindsight, it would have been better to reject the innovation early and move on to the next task. For this reason, testing processes in practice needs hard termination rules of the form: *if after drawing a certain number of samples no confident decision can be taken, then terminate the testing process rejecting the current innovation.* Denote by $\tau$ this capped early stopping rule and by $\mathrm{accept}$ the accept/reject decision rule that comes with it. We say that the pair $\pi = (\tau, \mathrm{accept})$ is a *policy* because it fully characterize the decision-making process for an innovation. Policies defined by capped early stopping rules (see (4) for a concrete example) are of great practical importance (Johari et al., 2017; Kohavi et al., 2013). However, policies can be defined more generally by any reasonable stopping rule and decision function. Given a (possibly infinite) set of policies, and assuming that $\mu_1, \mu_2, \ldots$ are drawn i.i.d. from some unknown but fixed distribution, the goal is to learn efficiently, at the lowest cost, the best policy $\pi_\star$ in the set with respect to a sensible metric. Competing against fixed policy classes is a common modeling choice that allows to express the intrinsic constraints that are imposed by the nature of the decision-making problem. For example, even if some policies outside of the class could theoretically yield better performance, they might not be implementable because of time, budget, fairness, or technology constraints.

**Challenges.** One of the biggest challenges arising in our framework is that running a decision-making policy generates a collection of samples that —in general— cannot be used to form an unbiased estimate of the policy reward (see the impossibility result in Section E of the Supplementary Material). The presence of this bias is a significant departure from settings like multiarmed and firing bandits (Auer et al., 2002; Jain and Jamieson, 2018), where the learner observes an unbiased sample of the target quantity at the end of every round (see the next section for additional details). Moreover, contrary to standard online learning problems, the relevant performance measure is neither additive in

the number of innovations nor in the number of samples per innovation. Therefore, algorithms have to be analyzed globally, and bandit-like techniques —in which the regret is additive over rounds— cannot be directly applied. We argue that these technical difficulties are a worthy price to pay in order to define a plausible setting, applicable to real-life scenarios.

**Main contributions.** The first contribution of this paper is providing a mathematical formalization of our ROI maximization setting for repeated decision making (Section 3). We then design an algorithm called Capped Policy Elimination (Algorithm 1, CAPE) that applies to finite policy classes (Section 4). We prove that CAPE converges to the optimal policy at rate $1/(\Delta^2 N)$, where $N$ is the number of tasks and $\Delta$ is the unknown gap between the performance of the two best policies, and at rate $N^{-1/3}$ when $\Delta$ is small (Theorem 1) . In Section 5 we tackle the challenging problem of infinitely large policy classes. For this setting, we design a preprocessing step (Algorithm 2, ESC) that leads to the ESC-CAPE algorithm. We prove that this algorithm converges to the optimal policy in an infinite set at a rate of $N^{-1/3}$ (Theorem 4).

**Limitations.** Although we do not investigate lower bounds in this paper, we conjecture that our $N^{-1/3}$ convergence rate it is optimal due to similarities with bandits with weakly observable feedback graphs (see Section 4, "Divided we fall"). Another limitation of our theory is that it only applies to i.i.d. sequences of values $\mu_n$. It would be interesting to extend our analysis to distributions of $\mu_n$ that evolve over time. These two intriguing problems are left open for future research.

## 2 Related Work

Return on Investment (ROI) was developed and popularized by Donaldson Brown in the early Nineties (Flesher and Previts, 2013) and it is still considered an extremely valuable metric by the overwhelming majority of marketing managers (Farris et al., 2010). Beyond economics, mathematics, and computer science, ROI finds applications in other fields, such as cognitive science and psychology (Chabris et al., 2009). Despite this, to the best of our knowledge, no theoretical online learning framework has been developed specifically for ROI maximization. However, our novel formalization of this sequential decision problem does share some similarities with other known online learning settings. In this section, we review the relevant literature regarding these settings and stress the differences with ours.

**Prophet inequalities and Pandora's box.** In prophet inequalities (Lucier, 2017; Correa et al., 2019; Alaei et al., 2012), an agent observes sequentially (usually non-negative) random variables $Z_1, \ldots, Z_n$ and decides to stop at some time $\tau$; the reward is then $Z_\tau$. Variants include the possibility of choosing more than one random variable (in which case the reward is some function of the selected random variables), and the possibility to go back in time (to some extent). The Pandora's box problem is slightly different (Weitzman, 1979; Kleinberg et al., 2016; Esfandiari et al., 2019); in its original formulation, the agent can pay a cost $c_n \geq 0$ to observe any $Z_n$. After stopping exploring, the agent's final utility is the maximum of the observed $Z_n$'s minus the cumulative cost (or, in other variants, some function of these). Similarly to the (general) prophet inequality, the agent in our sequential problem faces random variables ($Z_n = \mu_n$ in our notation) and sequentially selects any number of them (possibly with negative values) without the possibility to go back in time and change past decisions. The significant difference is that the agent in our setting never observes the value of $\mu_n$. In Pandora's box, the agent can see this value by paying some price (that approximately scales as $1/\varepsilon^2$ where $\varepsilon$ is the required precision). Finally, the global reward is the cumulative sum (as in prophets) and not the maximum (as in Pandora's box) of the selected variables, normalized by the total cost (as in Pandora's box, but our normalization is multiplicative instead of additive, as it represents a ROI).

**Multi-armed bandits.** If we think of the set of all policies used by the agent to determine whether or not to accept innovations as arms, our setting becomes somewhat reminiscent of multi-armed bandits (Slivkins, 2019; Bubeck and Cesa-Bianchi, 2012; Rosenberg et al., 2007). However, there are several notable differences between these two problems. In stochastic bandits, the agent observes an unbiased estimate of the expected reward of each pulled arm. In our setting, the agent not only does not see it directly, but it is mathematically impossible to define such an estimator solely with the feedback received (see the impossibility result in Section E of the Supplementary Material). Hence, off-the-shelf bandit algorithms cannot be run to solve our problem. In addition, the objective in bandits is to maximize the cumulative reward, which is additive over time, while the ROI is not. Thus, it is unclear how formal guarantees for bandit algorithms would translate to our problem.

We could also see firing bandits (Jain and Jamieson, 2018) as a variant of our problem, where $\mu_n$ belongs to $[0, 1]$, $\mathcal{D}_n$ are Bernoulli distribution with parameter $\mu_n$, and policies have a specific form that allows to easily define unbiased estimates of their rewards (which, we reiterate, is not possible in our setting in general). Furthermore, in firing bandits, it is possible to go back and forth in time, sampling from any of the past distributions $\mathcal{D}_n$ and gathering any number of samples from it. This is a reasonable assumption for the original motivations of firing bandits because the authors thought of $\mu_n$ as the value of a project in a crowdfunding platform, and, in their setting, drawing samples from $\mathcal{D}_n$ corresponds to displaying projects on web pages. However, in our setting, $\mu_n$ represents the theoretical increment (or decrement) of a company's profit through a given innovation, and it is unlikely that a company would show new interest in investing in a technology that has been tested before and did not prove to be useful (a killed project is seldom re-launched). Hence, when the sampling of $\mathcal{D}_n$ stops, an irrevocable decision is made. After that, the learner cannot draw any more samples in the future. Finally, as in multi-armed bandits, the performance criterion in firing bandits is the cumulative reward and not the global ROI.

Another online problem that shares some similarities with ours is bandits with knapsacks (Badanidiyuru et al., 2018). In this problem, playing an arm consumes one unit of time together with some other resources, and the learner receives an unbiased estimate of its reward as feedback. The process ends as soon as time or any one of the other resources is exhausted. As usual, the goal is to maximize the cumulative regret. As it turns out, we can also think of our problem as a budgeted problem. In this restatement, there is a budget of $T$ samples. The repeated decision-making process proceeds as before, but it stops as soon as the learner has drawn a total of $T$ samples across all decision tasks. The goal is again to maximize the total expected reward of accepted innovations divided by $T$ (see Section D of the Supplementary Material for more details on the reduction). As per the other bandit problems, there are two crucial differences. First, running a policy does not reveal an unbiased estimate of its reward. Second, our objective is different, and regret bounds do not directly imply convergence to optimal ROI. For other examples of budget-constrained bandit settings, see also (Cayci et al., 2020, 2019).

**Repeated A/B testing.**    We can view our problem as a framework for repeated A/B testing (Tukey, 1953; Genovese et al., 2006; Foster and Stine, 2008; Heesen and Janssen, 2016; Javanmard et al., 2018; Azevedo et al., 2018; Li and Barber, 2019; Schmit et al., 2019; Xu et al., 2021; Johari et al., 2021), in which assessing the value of an innovation corresponds to performing an A/B test, and the goal is maximizing the ROI. A popular metric to optimize sequential A/B tests is the so-called *false discovery rate* (FDR) —see (Ramdas et al., 2017; Yang et al., 2017) and references therein. Roughly speaking, the FDR is the ratio of accepted $\mu_n$ that are negative over the total number of accepted $\mu_n$ (or more generally, the number of incorrectly accepted tests over the total number if the metric used at each test changes with time). This, unfortunately, disregards the relative values of tests $\mu_n$ that must be taken into account when optimizing a single metric (Chen and Kasiviswanathan, 2020; Robertson and Wason, 2018). Indeed, the effect of many even slightly negative accepted tests could be overcome by a few largely positive ones. For instance, assume that the samples $X_{n,i}$ of any distribution $\mathcal{D}_n$ belong to $\{-1, 1\}$, and that their expected value $\mu_n$ is uniformly distributed on $\{-\varepsilon, \varepsilon\}$. To control the FDR, each A/B test should be run for approximately $1/\varepsilon^2$ times, yielding a ratio of the average value of an accepted test to the number of samples of order $\varepsilon^3$. A better strategy, using just one sample from each A/B test, is simply to accept $\mu_n$ if and only if the first sample is positive. Direct computations show that this policy, which fits our setting, achieves a significantly better performance of order $\varepsilon$.

Some other A/B testing settings are more closely related to ours, but make stronger additional assumptions or suppose preliminary knowledge: for example, smoothness assumptions can be made on both $\mathcal{D}_n$ and the distributions of $\mu_n$ (Azevedo et al., 2018), or the distribution of $\mu_n$ is known, and the distribution of samples belongs to a single parameter exponential family, also known beforehand (Schmit et al., 2019).

A related topic that sits in between multi-armed bandits and repeated A/B testing is best-arm identification—see, e.g., the recent paper (Garivier and Kaufmann, 2021) and references therein—where the learner has to minimize the number of observations needed to select a (near)-optimal arm.

**Rational metareasoning.**    Our setting is loosely related to the AI field of meta-reasoning (Griffiths et al., 2019; Hay et al., 2012). In a metalevel decision problem, determining the utility (or reward) of

a given action is computationally intractable. Instead, the learner can run a simulation, investing a computational cost to gather information about this hidden value. The high-level idea is then to learn *which* actions to simulate. After running some simulations, the learner picks an action to play, gains the corresponding (hidden) reward, and the state of the system changes. In rational meta-reasoning, the performance measure is the value of computation (VOC): the difference between the increment in expected utility gained by executing a simulation and the cost incurred by doing so. This setting is not directly comparable to ours for two reasons. First, the performance measure is different, and the additive nature of the difference that defines the VOC gives no guarantees on our multiplicative notion of ROI. Second, in this problem, one can pick which actions to simulate, while in our settings, innovations come independently of the learner, who has to evaluate them in that order.

## 3  Setting and Notation

In this section, we formally introduce the repeated decision-making protocol for an agent whose goal is to maximize the total return on investment in a sequence of decision tasks.

The only two choices that an agent makes in a decision task are when to stop gathering information on the current innovation and whether or not to accept the innovation based on this information. In other words, the behavior of the agent during each task is fully characterized by the choice of a pair $\pi = (\tau, \text{accept})$ that we call a *(decision-making) policy* (for the interested reader, Section A of the Supplementary Material contains a short mathematical discussion on policies), where:

- $\tau(\boldsymbol{x})$, called *duration*, maps a sequence of observations $\boldsymbol{x} = (x_1, x_2, \ldots)$ to an integer $d$ (the no. of observations after which the learner stops gathering info on the current innovation);

- $\text{accept}(d, \boldsymbol{x})$, called *decision*, maps the firs $d$ observations of a sequence $\boldsymbol{x} = (x_1, x_2, \ldots)$ to a boolean value in $\{0, 1\}$ (where 1 represents accepting the current innovation).

An instance of our repeated decision-making problem is therefore determined by a set of admissible policies $\Pi = \{\pi_k\}_{k \in \mathcal{K}} = \{(\tau_k, \text{accept})\}_{k \in \mathcal{K}}$ (with $\mathcal{K}$ finite or countable) and a distribution $\mathcal{D}$ on $[-1, 1]$, modelling the value of innovations.[1] Naturally, the former is known beforehand but the latter is unknown and should be learned.

For a fixed choice of $\Pi$ and $\mathcal{D}$, the protocol is formally described below. In each decision task $n$:

1. the *value* $\mu_n$ of the current innovation is drawn i.i.d. according to $\mathcal{D}$;
2. $\boldsymbol{X}_n$ is a sequence of i.i.d. (given $\mu_n$) *observations* with $X_{n,i} = \pm 1$ and $\mathbb{E}[X_{n,i} \mid \mu_n] = \mu_n$;
3. the agent picks $k_n \in \mathcal{K}$ or, equivalently, a policy $\pi_{k_n} = (\tau_{k_n}, \text{accept}) \in \Pi$;
4. the agent draws the first $d_n = \tau_{k_n}(\boldsymbol{X}_n)$ *samples*[2] of the sequence of observations $\boldsymbol{X}_n$;
5. on the basis of these sequential observations, the agent makes the decision $\text{accept}(d_n, \boldsymbol{X}_n)$.

Crucially, $\mu_n$ is *never* revealed to the learner. We say that the agent *runs a policy* $\pi_k = (\tau_k, \text{accept})$ (on a value $\mu_n$) when steps 4–5 occur (with $k_n \leftarrow k$). We also say that they accept (resp., rejects) $\mu_n$ if their decision at step 5 is equal to 1 (resp., 0). Moreover, we say that the *reward* obtained and the *cost*[3] paid by running a policy $\pi_k = (\tau_k, \text{accept})$ on a value $\mu_n$ are, respectively,

$$\text{reward}(\pi_k, \mu_n) = \mu_n \, \text{accept}\big(\tau_k(\boldsymbol{X}_n), \boldsymbol{X}_n\big) \in \{\mu_n, 0\} \qquad \text{cost}(\pi_k, \mu_n) = \tau_k(\boldsymbol{X}_n) \in \mathbb{N} \quad (1)$$

The objective of the agent is to converge to the highest ROI of a policy in $\Pi$, i.e., to guarantee that

$$R_N = \sup_{k \in \mathcal{K}} \frac{\sum_{n=1}^{N} \mathbb{E}\big[\text{reward}(\pi_k, \mu_n)\big]}{\sum_{m=1}^{N} \mathbb{E}\big[\text{cost}(\pi_k, \mu_m)\big]} - \frac{\sum_{n=1}^{N} \mathbb{E}\big[\text{reward}(\pi_{k_n}, \mu_n)\big]}{\sum_{m=1}^{N} \mathbb{E}\big[\text{cost}(\pi_{k_m}, \mu_m)\big]} \to 0 \quad \text{as } N \to \infty \quad (2)$$

where the expectations are taken with respect to $\mu_n$, $\boldsymbol{X}_n$, and (possibly) the random choices of $k_n$.

---

[1]We assume that the values of the innovations and the learner's observations belong to $[-1, 1]$ and $\{-1, 1\}$ respectively. We do this merely for the sake of readability (to avoid carrying over awkward constants or distributions $D_n$). With a standard argument, both $[-1, 1]$ and $\{-1, 1\}$ can be extended to arbitrary codomains straightforwardly under a mild assumption of subgaussianity.

[2]Given $\mu_n$, the random variable $d_n$ is a stopping time w.r.t. the natural filtration associated to $\boldsymbol{X}_n$.

[3]We define the cost as the duration $\tau_k(\boldsymbol{X}_n)$ of a run. Following our exact proofs, one can see that everything remains true if we define the cost in terms of *any* increasing function of the duration. This puts no meaningful restrictions on cost functions.

To further lighten notations, we denote the expected reward, cost, and ROI of a policy $\pi$ by

$$\text{reward}(\pi) = \mathbb{E}\big[\text{reward}(\pi, \mu_n)\big], \ \text{cost}(\pi) = \mathbb{E}\big[\text{cost}(\pi, \mu_n)\big], \ \text{ROI}(\pi) = \text{reward}(\pi)/\text{cost}(\pi) \quad (3)$$

respectively and we say that $\pi_{k^\star}$ is an *optimal policy* if $k^\star \in \text{argmax}_{k \in \mathcal{K}} \text{ROI}(\pi_k)$. Note that reward$(\pi)$ and cost$(\pi)$ do not depend on $n$ because $\mu_n$ is drawn i.i.d. according to $\mu$.

For each policy $(\tau, \text{accept}) \in \Pi$ and all tasks $n$, we allow the agent to reject the value $\mu_n$ regardless of the outcome of the sampling. Formally, the agent can always run the policy $(\tau, 0)$, where the second component of the pair is the decision identically equal to zero (i.e., the rule "always reject").

We also allow the agent to draw arbitrarily many extra samples in addition to the number $\tau(\boldsymbol{X}_n)$ that they would otherwise draw when running a policy $(\tau, \text{accept}) \in \Pi$ on a value $\mu_n$, provided that these additional samples are not taken into account in the decision to either accept or reject $\mu_n$. Formally, the agent can always draw $\tau(\boldsymbol{X}_n) + k$ many samples (for any $k \in \mathbb{N}$) before making the decision $\text{accept}\big(\tau(\boldsymbol{X}_n), \boldsymbol{X}_n\big)$, where we stress that the first argument of the decision function accept is $\tau(\boldsymbol{X}_n)$ and not $\tau(\boldsymbol{X}_n) + k$. Oversampling this way worsens the objective and might seem utterly counterproductive, but it will be crucial for recovering unbiased estimates of $\mu_n$.

## 4    Competing Against $K$ policies (CAPE)

As we mentioned in the introduction, in practice the duration of a decision task is defined by a capped early-stopping rule —e.g., drawing samples until $0$ falls outside of a confidence interval around the empirical average, or a maximum number of draws has been reached. More precisely, if $N$ tasks have to be performed, one could consider the natural policy class $\{(\tau_k, \text{accept})\}_{k \in \{1,\ldots,K\}}$ given by

$$\tau_k(\boldsymbol{x}) = \min\big(k, \ \inf\{d \in \mathbb{N} : |\overline{x}_d| \geq \alpha_d\}\big) \qquad \text{and} \qquad \text{accept}(d, \boldsymbol{x}) = \mathbb{I}\{\overline{x}_d \geq \alpha_d\} \quad (4)$$

where $\overline{x}_d = (1/d) \sum_{i=1}^d x_i$ is the average of the first $d$ elements of the sequence $\boldsymbol{x} = (x_1, x_2, \ldots)$ and $\alpha_d = c\sqrt{(1/d)\ln(KN/\delta)}$, for some $c > 0$ and $\delta \in (0,1)$. While in this example policies are based on an Hoeffding concentration rule, in principle the learner is free to follow any scheme. Thus, we now generalize this notion and present an algorithm with provable guarantees against these finite families of policies.

**Finite sets of policies.**    In this section, we focus on finite sets of $K$ policies $\Pi = \{\pi_k\}_{k \in \{1,\ldots,K\}} = \{(\tau_k, \text{accept})\}_{k \in \{1,\ldots,K\}}$ where accept is an arbitrary decision and $\tau_1, \ldots, \tau_K$ is any sequence of bounded durations (say, $\tau_k \leq k$ for all $k$).[4] For the sake of convenience, we assume the durations are sorted by index ($\tau_k \leq \tau_h$ if $k \leq h$), so that $\tau_1$ is the shortest and $\tau_K$ is the longest.

**Divided we fall.**    A common strategy in online learning problems with limited feedback is explore-then-commit (ETC). ETC consists of two phases. In the first phase (explore), each action is played for the same amount of rounds, collecting this way i.i.d. samples of all rewards. In the subsequent commit phase, the arm with the best empirical observations is played consistently. Being very easy to execute, this strategy is popular in practice, but unfortunately, it is theoretically suboptimal in some applications. A better approach is performing action elimination. In a typical implementation of this strategy, all actions in a set are played with a round-robin schedule, collecting i.i.d. samples of their rewards. At the end of each cycle, all actions that are deemed suboptimal are removed from the set, and a new cycle begins. Neither one of these strategies can be applied directly because running a policy in our setting does not return an unbiased estimate of its reward (for a quick proof of this simple result, see Section E in the Supplementary Material). However, it turns out that we can get an i.i.d. estimate of a policy $\pi$ by playing a *different* policy $\pi'$. Namely, one that draws *more* samples than $\pi$. This is reminiscent of bandits with a weakly observable feedback graph, a related problem for which the time-averaged regret over $T$ rounds vanishes at a $T^{-1/3}$ rate (Alon et al., 2015). Albeit none of these three techniques works on its own, suitably interweaving all of them does.

**United we stand.**    With this in mind, we now present our simple and efficient algorithm (Algorithm 1, CAPE) whose ROI converges (with high probability) to the best one in a finite family of policies. We will later discuss how to extend the analysis even further, including countable families of policies. Our algorithm performs policy elimination (lines 1–5) for a certain number of tasks

---

[4]We chose $\tau_k \leq k$ for the sake of concreteness. All our results can be straightforwardly extended to arbitrary $\tau_k \leq D_k$ by simply assuming without loss of generality that $k \mapsto D_k$ is monotone and replacing $k$ with $D_k$.

(line 1) or until a single policy is left (line 6). After that, it runs the best policy left in the set (line 7), breaking ties arbitrarily,[5] for all remaining tasks. During each policy elimination step, the algorithm oversamples (line 2) by drawing twice as many samples as it would suffice to take its decision $\text{accept}\big(\tau_{\max(C_n)}(\boldsymbol{X}_n), \boldsymbol{X}_n\big)$ (at line 3). These extra samples are used to compute rough estimates of rewards and costs of all potentially optimal policies and more specifically to build *unbiased* estimates of these rewards. The test at line 4 has the only purpose of ensuring that the denominators $\widehat{c}_n^-(k)$ at line 5 are bounded away from zero so that all quantities are well-defined.

---

**Algorithm 1:** Capped Policy Elimination (CAPE)

---

**Input:** finite policy set $\Pi$, number of tasks $N$, confidence parameter $\delta$, exploration cap $N_{\text{ex}}$
**Initialization:** let $C_1 \leftarrow \{1, \ldots, K\}$ be the set of indices of all currently optimal candidates

1    **for** *task* $n = 1, \ldots, N_{\text{ex}}$ **do**
2        draw the first $2\max(C_n)$ samples $X_{n,1}, \ldots, X_{n,2\max(C_n)}$ of $\boldsymbol{X}_n$
3        make the decision $\text{accept}\big(\tau_{\max(C_n)}(\boldsymbol{X}_n), \boldsymbol{X}_n\big)$
4        **if** $n \geq 2K^2 \ln(4KN_{\text{ex}}/\delta)$ **then** let $C_{n+1} \leftarrow C_n \setminus C_n'$, where

5
$$C_n' = \big\{ k \in C_n \,:\, \big(\widehat{r}_n^+(k) \geq 0 \text{ and } \widehat{r}_n^+(k)/\widehat{c}_n^-(k) < \widehat{r}_n^-(j)/\widehat{c}_n^+(j), \text{ for some } j \in C_n\big)$$
$$\text{or } \big(\widehat{r}_n^+(k) < 0 \text{ and } \widehat{r}_n^+(k)/\widehat{c}_n^+(k) < \widehat{r}_n^-(j)/\widehat{c}_n^-(j), \text{ for some } j \in C_n\big)\big\}$$

$$\widehat{r}_n^{\pm}(k) = \frac{1}{n}\sum_{m=1}^{n}\sum_{i=1}^{\max(C_m)}\frac{X_{m,\max(C_m)+i}}{\max(C_m)}\,\text{accept}\big(\tau_k(\boldsymbol{X}_m), \boldsymbol{X}_m\big) \pm \sqrt{\frac{2}{n}\ln\frac{4KN_{\text{ex}}}{\delta}} \quad (5)$$

$$\widehat{c}_n^{\pm}(k) = \frac{1}{n}\sum_{m=1}^{n}\tau_k(\boldsymbol{X}_m) \pm (k-1)\sqrt{\frac{1}{2n}\ln\frac{4KN_{\text{ex}}}{\delta}} \quad\quad\quad (6)$$

6        **if** $|C_{n+1}| = 1$ **then** let $\widehat{r}_{N_{\text{ex}}}^{\pm}(k) \leftarrow \widehat{r}_n^{\pm}(k)$, $\widehat{c}_{N_{\text{ex}}}^{\pm}(k) \leftarrow \widehat{c}_n^{\pm}(k)$, $C_{N_{\text{ex}}+1} \leftarrow C_{n+1}$, **break**
7    run policy $\pi_{k'}$ for all remaining tasks, where

$$k' \in \begin{cases} \underset{k \in C_{N_{\text{ex}}+1}}{\text{argmax}}\ \big(\widehat{r}_{N_{\text{ex}}}^+(k)/\widehat{c}_{N_{\text{ex}}}^-(k)\big) & \text{if } \widehat{r}_{N_{\text{ex}}}^+(k) \geq 0 \text{ for some } k \in C_{N_{\text{ex}}+1} \\ \underset{k \in C_{N_{\text{ex}}+1}}{\text{argmax}}\ \big(\widehat{r}_{N_{\text{ex}}}^+(k)/\widehat{c}_{N_{\text{ex}}}^+(k)\big) & \text{if } \widehat{r}_{N_{\text{ex}}}^+(k) < 0 \text{ for all } k \in C_{N_{\text{ex}}+1} \end{cases} \quad (7)$$

---

As usual in online learning, the *gap* in performance between optimal and sub-optimal policies is a complexity parameter. We define it as $\Delta = \min_{k \neq k^\star}\big(\text{ROI}(\pi_{k^\star}) - \text{ROI}(\pi_k)\big)$, where we recall that $k^\star \in \text{argmax}_k \text{ROI}(\pi_k)$ is the index of an optimal policy. Conventionally, we set $1/0 = \infty$.

**Theorem 1.** *If $\Pi$ is a finite set of $K$ policies, then the ROI of Algorithm 1 run for $N$ tasks with exploration cap $N_{\text{ex}} = \lceil N^{2/3} \rceil$ and confidence parameter $\delta \in (0,1)$ converges to the optimal* $\text{ROI}(\pi_{k^\star})$*, with probability at least $1 - \delta$, at a rate*

$$R_N = \widetilde{\mathcal{O}}\left(\min\left(\frac{K^3}{\Delta^2 N}, \frac{K}{N^{1/3}}\right)\right)$$

*as soon as $N \geq K^3$ (where the $\widetilde{\mathcal{O}}$ notation hides only logarithmic terms, including a $\log(1/\delta)$ term).*

*Proof sketch.* This theorem relies on four technical lemmas (Lemmas 5-8) whose proofs are deferred to Section B of the Supplementary Material.

With a concentration argument (Lemma 5), we leverage the definitions of $\widehat{r}_n^{\pm}(k), \widehat{c}_n^{\pm}(k)$ and the i.i.d. assumptions on the samples $X_{n,i}$ to show that, with probability at least $1 - \delta$, the event

$$\widehat{r}_n^-(k) \leq \text{reward}(\pi_k) \leq \widehat{r}_n^+(k) \quad\quad \text{and} \quad\quad \widehat{c}_n^-(k) \leq \text{cost}(\pi_k) \leq \widehat{c}_n^+(k) \quad (8)$$

---

[5]More precisely, ties should be broken in a measurable way, e.g., uniformly at random. We do not insist on this point here, but the interested reader might see (Cesari and Colomboni, 2021, Section 2.4) for a more thorough discussion on this topic.

occurs simultaneously for all $n \le N_{\text{ex}}$ and all $k \le \max(C_n)$. For the rewards, the key is oversampling, because $\text{accept}\big(\tau_k(\boldsymbol{X}_m), \boldsymbol{X}_m\big)$ in eq. (5) depends only on the first $k \le \max(C_m)$ samples of $\boldsymbol{X}_m$ and is therefore independent of $X_{m,\max(C_m)+i}$ for all $i$. Assume now that (8) holds.

If $\Delta > 0$ (i.e., if there is a unique optimal policy), we then obtain (Lemma 6) that suboptimal policies are eliminated after at most $N'_{\text{ex}}$ tasks, where $N'_{\text{ex}} \le 288 \, K^2 \ln(4KN_{\text{ex}}/\delta)/\Delta^2 + 1$. To prove it we show that a confidence interval for $\text{ROI}(\pi_k) = \text{reward}(\pi_k)/\text{cost}(\pi_k)$ is given by

$$\left[ \frac{\widehat{r}_n^-(k)}{\widehat{c}_n^+(k)} \mathbb{I}\{\widehat{r}_n^+(k) \ge 0\} + \frac{\widehat{r}_n^-(k)}{\widehat{c}_n^-(k)} \mathbb{I}\{\widehat{r}_n^+(k) < 0\}, \ \frac{\widehat{r}_n^+(k)}{\widehat{c}_n^-(k)} \mathbb{I}\{\widehat{r}_n^+(k) \ge 0\} + \frac{\widehat{r}_n^+(k)}{\widehat{c}_n^+(k)} \mathbb{I}\{\widehat{r}_n^+(k) < 0\} \right]$$

we upper bound its length, and we compute an $N'_{\text{ex}}$ such that this upper bound is smaller than $\Delta/2$.

Afterwards, we analyze separately the case in which the test at line 6 is true for some task $N'_{\text{ex}} \le N_{\text{ex}}$ and its complement (i.e., when the test is always false).

In the first case, by (8) there exists a unique optimal policy, i.e., we have that $\Delta > 0$. This is where the policy-elimination analysis comes into play. We can apply the bound above on $N'_{\text{ex}}$, obtaining a deterministic upper bound $N''_{\text{ex}}$ on the number $N'_{\text{ex}}$ of tasks needed to identify the optimal policy. Using this upper bound, writing the definition of $R_N$, and further upper bounding (Lemma 7) yields

$$R_N \le \min\left( \frac{(2K+1)N_{\text{ex}}}{N}, \ \frac{(2K+1)\big(288\,(K/\Delta)^2 \ln(4KN_{\text{ex}}/\delta) + 1\big)}{N} \right) \tag{9}$$

Finally, we consider the case in which the test at line 6 is false for all tasks $n \le N_{\text{ex}}$, and line 7 is executed with $C_{N_{\text{ex}}+1}$ containing two or more policies. This is covered by a worst case explore-then-commit analysis. The key idea here is to use the definition of $k'$ in Equation (7) to lower bound $\text{reward}(\pi_{k'})$ in terms of $\text{reward}(\pi_{k^\star})/\text{cost}(\pi_{k^\star})$. This, together with some additional technical estimations (Lemma 8) leads to the result. $\qquad \square$

As we mentioned in Footnote 4, if we swap the bounds on durations $\tau_k \le k$ with generic $\tau_k \le D_k$ (with $k \mapsto D_k$ non-decreasing), the result would still hold, but the right-hand side would scale with $D_K$ rather than $K$. To see why, note that in our current presentation when the policy set $\Pi$ is finite, $K$ plays the role of both the cardinality of $\Pi$ and a uniform upper bound on their durations, where the latter role is the most important. The key intuition on why this is the case is that whenever some data are sufficient to evaluate a policy $k$, it can also be used to evaluate all policies $k' \le k$ for free. In particular, in a case where many (even infinitely many, in principle) of the durations of the policies shared the same upper bound $D_k$, our algorithm would have a very easy time in narrowing down the best one because, using only $D_k$ samples, it could keep updated a large number of policy estimators all at once.

## 5  Competing Against Infinitely Many Policies (ESC-CAPE)

Theorem 1 provides theoretical guarantees on the convergence rate $R_N$ of CAPE to the best ROI of a finite set of policies. Unfortunately, the bound becomes vacuous when the cardinality $K$ of the policy set is large compared to the number of tasks $N$. It is therefore natural to investigate whether the problem becomes impossible in this scenario.

**Infinite sets of policies.**  With this goal in mind, we now focus on policy sets $\Pi = \{\pi_k\}_{k \in \mathcal{K}} = \big\{(\tau_k, \text{accept})\big\}_{k \in \mathcal{K}}$ as in the previous section, with $\mathcal{K} = \mathbb{N}$ rather than $\mathcal{K} = \{1, \ldots, K\}$.

We will show how such a countable set of policies can be reduced to a finite one containing all optimal policies with high probability (Algorithm 2, ESC). After this is done, we can run CAPE on the smaller policy set, obtaining theoretical guarantees for the resulting algorithm.

**Estimating rewards and costs.**  Similarly to eqs. (5) and (6), we first introduce estimators for our target quantities. If at least $2k$ samples are drawn during each of $n_2$ consecutive tasks $n_1 + 1, \ldots, n_1 + n_2$, we can define, for all $\varepsilon > 0$, the following lower confidence bound on $\text{reward}(\pi_k)$:

$$\widehat{r}_k^-(n_1, n_2, \varepsilon) = \frac{1}{n_2} \sum_{n=n_1+1}^{n_1+n_2} \sum_{i=1}^{k} \frac{X_{n,k+i}}{k} \text{accept}\big(\tau_k(\boldsymbol{X}_n), \boldsymbol{X}_n\big) - 2\varepsilon \tag{10}$$

If at least $\tau_k(\boldsymbol{X}_n)$ samples are drawn during each of $m_0$ consecutive tasks $n_0 + 1, \ldots, n_0 + m_0$, we can define the following empirical average of $\text{cost}(\pi_k)$:

$$\bar{c}_k(n_0, m_0) = \big(\tau_k(\boldsymbol{X}_{n_0+1}) + \ldots + \tau_k(\boldsymbol{X}_{n_0+m_0})\big)/m_0 \tag{11}$$

**A key observation.** The key idea behind Algorithm 2 (ESC) is simple. Since all optimal policies $\pi_{k^\star}$ have to satisfy the relationships $\mathrm{reward}(\pi_k)/\mathrm{cost}(\pi_k) \leq \mathrm{reward}(\pi_{k^\star})/\mathrm{cost}(\pi_{k^\star}) \leq 1/\mathrm{cost}(\pi_{k^\star})$, then, for all policies $\pi_k$ with positive $\mathrm{reward}(\pi_k)$, the cost of any optimal policy $\pi_{k^\star}$ must satisfy the relationship $\mathrm{cost}(\pi_{k^\star}) \leq \mathrm{cost}(\pi_k)/\mathrm{reward}(\pi_k)$. In other words, *optimal policies cannot draw too many samples* and their cost can be controlled by estimating the reward and cost of *any* policy with positive reward.

We recall that running a policy $(\tau, 0)$ during a task $n$ means drawing the first $\tau(\boldsymbol{X}_n)$ samples of $\boldsymbol{X}_n = (X_{n,1}, X_{n,2}, \ldots)$ and always rejecting $\mu_n$, regardless of the observations.

---

**Algorithm 2:** Extension to Countable (ESC)

---

**Input:** countable policy set $\Pi$, number of tasks $N$, confidence parameter $\delta$, accuracy levels $(\varepsilon_n)_n$
**Initialization:** for all $j$, let $m_j \leftarrow \left\lceil \ln\big(j(j+1)/\delta\big)/2\varepsilon_j^2 \right\rceil$ and $M_j = m_1 + \ldots + m_j$

1   **for** $j = 1, 2, \ldots$ **do**
2      run policy $\big(2 \cdot 2^j, 0\big)$ for $m_j$ tasks and compute $\widehat{r}_{2^j}^- \leftarrow \widehat{r}_{2^j}^-(M_{j-1}, m_j, \varepsilon_j)$ as in (10)
3      **if** $\widehat{r}_{2^j}^- > 0$ **then** let $j_0 \leftarrow j$ and $k_0 \leftarrow 2^{j_0}$
4         **for** $l = j_0 + 1, j_0 + 2, \ldots$ **do**
5            run policy $\big(\tau_{2^l}, 0\big)$ for $m_l$ tasks and compute $\overline{c}_{2^l} \leftarrow \overline{c}_{2^l}(M_{l-1}, m_l)$ as in (11)
6            **if** $\overline{c}_{2^l} > 2^l \varepsilon_l + k_0/\widehat{r}_{k_0}^-$ **then** let $j_1 \leftarrow l$ and **return** $K \leftarrow 2^{j_1}$

---

Thus, Algorithm 2 (ESC) first finds a policy $\pi_{k_0}$ with $\mathrm{reward}(\pi_{k_0}) > 0$ (lines 1–3), memorizing an upper estimate $k_0/\widehat{r}_{k_0}^-$ of the ratio $\mathrm{cost}(\pi_{k_0})/\mathrm{reward}(\pi_{k_0}) = 1/\mathrm{ROI}(\pi_{k_0})$. By the argument above, this estimate upper bounds the expected number of samples $\mathrm{cost}(\pi_{k^\star})$ drawn by *all* optimal policies $\pi_{k^\star}$. Then ESC simply proceeds to finding the smallest (up to a factor of 2) $K$ such that $\mathrm{cost}(\pi_K) \geq k_0/\widehat{r}_{k_0}^-$ (lines 4–6). Being $k_0/\widehat{r}_{k_0}^- \geq \mathrm{cost}(\pi_{k_0})/\mathrm{reward}(\pi_{k_0}) \geq \mathrm{cost}(\pi_{k^\star})$ by construction, the index $K$ determined this way upper bounds $k^\star$ for all optimal policies $\pi_{k^\star}$. (All the previous statements are intended to hold with high probability.) This is formalized in the following key lemma, whose full proof we defer to Section C of the Supplementary Material.

**Lemma 2.** *Let $\Pi$ be a countable set of policies. If ESC is run with $\delta \in (0, 1)$, $\varepsilon_1, \varepsilon_2, \ldots \in (0, 1]$, and halts returning $K$, then $k^\star \leq K$ for all optimal policies $\pi_{k^\star}$ with probability at least $1 - \delta$.*

Before proceeding with the main result of this section, we need a final lemma upper bounding the expected cost of our ESC algorithm. This step is crucial to control the total ROI because in this setting with arbitrarily long durations, picking the wrong policy even once is, in general, enough to drop the performance of an algorithm down to essentially zero, compromising the convergence to an optimal policy. This is another striking difference with other common online learning settings like stochastic bandits, where a single round has a negligible influence on the overall performance of an algorithm. To circumvent this issue, we designed ESC so that it tests shorter durations first, stopping as soon as the previous lemma applies, and a finite upper bound $K$ on $k^\star$ is determined.

**Lemma 3.** *Let $\Pi$ be a countable set of policies. If ESC is run with $\delta \in (0, 1)$, $\varepsilon_1, \varepsilon_2, \ldots \in (0, 1]$, and halts returning $K$, then the total number of samples it draws before stopping (i.e., its cost) is upper bounded by $\widetilde{\mathcal{O}}\big((K/\varepsilon^2) \log(1/\delta)\big)$, where $\varepsilon = \min\{\varepsilon_1, \varepsilon_2, \ldots, \varepsilon_{\log_2 K}\}$.*

*Proof.* Note that, by definition, $\varepsilon = \min\{\varepsilon_1, \varepsilon_2, \ldots, \varepsilon_{j_1}\} > 0$. Algorithm 2 (ESC) draw samples only when lines 2 or 5 are executed. Whenever line 2 is executed ($j = 1, \ldots, j_0$) the algorithm performs $m_j$ tasks drawing $2 \cdot 2^j$ samples each time. Similarly, whenever line 5 is executed ($l = j_0 + 1, \ldots, j_1$) the algorithm draws at most $2^l$ samples during each of the $m_l$ tasks. Therefore, recalling that $j_1 = \log_2 K$, the total number of samples drawn by ESC before stopping is at most

$$\sum_{j=1}^{j_0} 2 \cdot 2^j m_j + \sum_{l=j_0+1}^{j_1} 2^l m_l \leq 2 \sum_{j=1}^{j_1} 2^j m_j \leq 2 j_1 2^{j_1} \left\lceil \frac{1}{2\varepsilon^2} \ln \frac{j_1(j_1+1)}{\delta} \right\rceil \qquad \square$$

**The ESC-CAPE algorithm.** We can now join together our two algorithms obtaining a new one, that we call ESC-CAPE, which takes as input a countable policy set $\Pi$, the number of tasks $N$, a confidence parameter $\delta$, some accuracy levels $\varepsilon_1, \varepsilon_2, \ldots$, and an exploration cap $N_{\mathrm{ex}}$. The joint

algorithm runs ESC first with parameters $\Pi, N, \delta, \varepsilon_1, \varepsilon_2, \dots$. Then, if ESC halts returning $K$, it runs CAPE with parameters $\{(\tau_k, \text{accept})\}_{k \in \{1,\dots,K\}}, N, \delta, N_{\text{ex}}$.

**Analysis of ESC-CAPE.**  Since ESC rejects all values $\mu_n$, the sum of the rewards accumulated during its run is zero. Thus, the only effect that ESC has on the convergence rate $R_N$ of ESC-CAPE is an increment on the total cost in the denominator of its ROI. We control this cost by minimizing its upper bound in Lemma 3. This is not a simple matter of taking all $\varepsilon_j$'s as large as possible. Indeed, if all the $\varepsilon_j$'s are large, the **if** clause at line 3 might never be verified, because the conditional checks whether a confidence interval (for $\text{reward}(\pi_{2^j})$) of length $\Theta(\varepsilon)$ sits to the right of the origin. In other words, the returned index $K$ depends on $\varepsilon$ and grows unbounded in general as $\varepsilon$ approaches $1/2$. This follows directly from the definition of our lower estimate on the rewards (10). Thus, there is a trade-off between having a small $K$ (which requires small $\varepsilon_j$'s) and a small $1/\varepsilon^2$ to control the cost of ESC (for which we need large $\varepsilon_j$'s). A direct computation shows that picking constant accuracy levels $\varepsilon_j = N^{-1/3}$ for all $j$ achieves the best of both worlds and immediately gives our final result.

**Theorem 4.** *If $\Pi$ is a countable set of policies, then the ROI of ESC-CAPE run for $N$ tasks with confidence parameter $\delta \in (0,1)$, constant accuracy levels $\varepsilon_j = N^{-1/3}$, and exploration cap $N_{\text{ex}} = \lceil N^{2/3} \rceil$ converges to the optimal $\text{ROI}(\pi_{k^\star})$, with probability at least $1 - \delta$, at a rate*

$$R_N = \widetilde{\mathcal{O}} \left( \frac{1 + K \mathbb{I}\{\text{ESC halts returning } K\}}{N^{1/3}} \right)$$

*where the $\widetilde{\mathcal{O}}$ notation hides only logarithmic terms, including a $\log(1/\delta)$ term.*

Note that the previous bound depends on the algorithm-dependent quantity $K$. While this might seem somewhat nonconventional, a simple argument gets rid of this issue. Indeed, if a lower bound $\lambda^\star$ on the ROI of an optimal policy were known beforehand, then it would be easy to control $K$ by estimating the cost of policies $2^\ell$, for increasing values of $\ell$ (as in ESC). In this case, $K$ would be upper bounded by the number of policies whose cost is smaller than $2/\lambda^\star$ (say). This quantity is now constant (in $N$; it depends only on the distribution $\mathcal{D}$ and the set of policies $\Pi$, playing the role of a complexity measure of the problem instance). If $\lambda^\star$ is unknown, one can first estimate it up to some multiplicative constant (say, $1/2$), with high probability. Again, we can do this at a constant cost (depending only on the instance $(\Pi, \mathcal{D})$) via a multiplicative Chernoff bound.

## 6 Conclusions

After formalizing the problem of ROI maximization in repeated decision making, we presented an algorithm (ESC-CAPE) that is competitive against infinitely large policy sets (Theorem 4). For this algorithm, we prove a convergence rate of order $1/N^{1/3}$ with high probability. To analyze it, we first proved a convergence result for its finite counterpart CAPE (Theorem 1), which is of independent interest. Notably, this finite analysis guarantees a significantly faster convergence of order $1/N$ on easier instances in which there is a positive gap in performance between the two best policies.

## Acknowledgments and Disclosure of Funding

An earlier version of this work was done during Tommaso Cesari's Ph.D. at the University of Milan. Nicolò Cesa-Bianchi and Tommaso R. Cesari gratefully acknowledge partial support by Criteo AI Lab through a Faculty Research Award and by the MIUR PRIN grant Algorithms, Games, and Digital Markets (ALGADIMAR). Nicolò Cesa-Bianchi was also supported by the COST Action CA16228 "European Network for Game Theory" (GAMENET) and by the EU Horizon 2020 ICT-48 research and innovation action under grant agreement 951847, project ELISE. This work has also benefited from the AI Interdisciplinary Institute ANITI. ANITI is funded by the French "Investing for the Future – PIA3" program under the Grant agreement n. ANR-19-PI3A-0004. Yishay Mansour was supported in part by a grant from the European Research Council (ERC) under the European Union'sHorizon 2020 research and innovation program (grant agreement No. 882396), by the Israel Science Foundation(grant number 993/17), Tel Aviv University Center for AI and Data Science (TAD), and the Yandex Initiative for Machine Learning at Tel Aviv University. Vianney Perchet was supported by a public grant as part of the Investissement d'avenir project, reference ANR-11-LABX-0056-LMH, LabEx LMH, in a joint call with Gaspard Monge Program for optimization, operations research and their interactions with data sciences. Vianney Perchet also acknowledges the support of the ANR under the grant ANR-19-CE23-0026.

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
