# ROI Maximization
# in Stochastic Online Decision-Making
## Supplementary Material

## A    Decision-Making Policies

In this section, we give a formal functional definition of the decision-making policies introduced in Section 3. During each task, the agent sequentially observes samples $x_i \in [-1, 1]$ representing realizations of stochastic observations of the current innovation value. A map $\tau \colon [-1, 1]^{\mathbb{N}} \to \mathbb{N}$ is a *duration* (of a decision task) if for all $\boldsymbol{x} \in [-1, 1]^{\mathbb{N}}$, its value $d = \tau(\boldsymbol{x}) \in \mathbb{N}$ at $\boldsymbol{x}$ depends only on the first $d$ components $x_1, x_2, \ldots, x_d$ of $\boldsymbol{x} = (x_1, x_2, \ldots)$; mathematically speaking, if $\boldsymbol{X}$ is a discrete stochastic process (i.e., a random sequence), then $\tau(\boldsymbol{X})$ is a stopping time with respect to the filtration generated by $\boldsymbol{X}$. This definition reflects the fact that the components $x_1, x_2, \ldots$ of the sequence $\boldsymbol{x} = (x_1, x_2, \ldots)$ are generated sequentially, and the decision to stop testing an innovation depends only on what occurred so far. A concrete example of a duration function is the one, mentioned in the introduction and formalized in (4), that keeps drawing samples until the empirical average of the observed values $x_i$ surpasses/falls below a certain threshold, or a maximum number of samples have been drawn.

To conclude a task, the agent has to make a decision: either accepting or rejecting the current innovation. Formally, we say that a function $\mathrm{accept} \colon \mathbb{N} \times [-1, 1]^{\mathbb{N}} \to \{0, 1\}$ is a *decision* (to accept) if for all $d \in \mathbb{N}$ and $\boldsymbol{x} \in [-1, 1]^{\mathbb{N}}$, its value $\mathrm{accept}(d, \boldsymbol{x}) \in \{0, 1\}$ at $(d, \boldsymbol{x})$ depends only on the first $d$ components $x_1, \ldots, x_d$ of $\boldsymbol{x} = (x_1, x_2, \ldots)$. Again, this definition reflects the fact that the decision $\mathrm{accept}(d, \boldsymbol{x})$ to either accept ($\mathrm{accept}(d, \boldsymbol{x}) = 1$) or reject ($\mathrm{accept}(d, \boldsymbol{x}) = 0$) the current innovation after observing the first $d$ values $x_1, \ldots, x_d$ of $\boldsymbol{x} = (x_1, x_2, \ldots)$ is oblivious to all future observations $x_{d+1}, x_{d+2}, \ldots$. Following up on the concrete example above, the decision function is accepting the current innovation if and only if the the empirical average of the observed values $x_i$ surpasses a certain threshold.[6]

Since the only two choices that an agent makes in a decision task are when to stop drawing new samples and whether or not to accept the current innovation, the behavior of the agent during each task is fully characterized by the choice of a pair $\pi = (\tau, \mathrm{accept})$ that we call a (*decision-making*) *policy*, where $\tau$ is a duration and $\mathrm{accept}$ is a decision.

## B    Technical Lemmas for Theorem 1

In this section, we give formal proofs of all results needed to prove Theorem 1.

**Lemma 5.** *Under the assumptions of Theorem 1, the event*

$$\widehat{r}_n^-(k) \le \mathrm{reward}(\pi_k) \le \widehat{r}_n^+(k) \qquad and \qquad \widehat{c}_n^-(k) \le \mathrm{cost}(\pi_k) \le \widehat{c}_n^+(k) \tag{12}$$

*occurs simultaneously for all $n = 1, \ldots, N_{\mathrm{ex}}$ and all $k = 1, \ldots, \max(C_n)$ with probability at least $1 - \delta$.*

*Proof.* Let, for all $n, k$,

$$\varepsilon_n = \sqrt{\frac{\ln(4KN_{\mathrm{ex}}/\delta)}{2n}}, \qquad \overline{r}_n(k) = \widehat{r}_n^+(k) - 2\varepsilon_n, \qquad \overline{c}_n(k) = \widehat{c}_n^+(k) - (k-1)\varepsilon_n \tag{13}$$

---

[6]Note that, even for decision functions that only look at the mean of the first $d$ values, our definition is more general than simple threshold functions of the form $\mathbb{I}\{\mathrm{mean} \ge \varepsilon_d\}$, as it also includes all decisions of the form $\mathbb{I}\{\mathrm{mean} \in A_d\}$, for all measurable $A_d \subset \mathbb{R}$.

Note that $\bar{c}_n(k)$ is the empirical average of $n$ i.i.d. samples of $\mathrm{cost}(\pi_k)$ for all $n, k$ by definitions (13), (6), (1), (3), and point 4 in the formal definition of our protocol (Section 3). We show now that $\bar{r}_n(k)$ is the empirical average of $n$ i.i.d. samples of $\mathrm{reward}(\pi_k)$ for all $n, k$; then claim (8) follows by Hoeffding's inequality. Indeed, by the conditional independence of the samples and being $\mathrm{accept}(k, \boldsymbol{x})$ independent of the variables $(x_{k+1}, x_{k+2}, \dots)$ by definition, for all tasks $n$, all policies $k \in C_n$, and all $i > \max(C_n)$ ($\geq k$ by monotonicity of $k \mapsto k$),

$$\mathbb{E}\Big[X_{n,i}\,\mathrm{accept}\big(\tau_k(\boldsymbol{X}_n), \boldsymbol{X}_n\big)\,\Big|\,\mu_n\Big] = \mathbb{E}\left[X_{n,i}\mid\mu_n\right]\mathbb{E}\Big[\mathrm{accept}\big(\tau_k(\boldsymbol{X}_n), \boldsymbol{X}_n\big)\,\Big|\,\mu_n\Big]$$

$$= \mu_n\,\mathbb{E}\Big[\mathrm{accept}\big(\tau_k(\boldsymbol{X}_n), \boldsymbol{X}_n\big)\,\Big|\,\mu_n\Big]$$

$$= \mathbb{E}\Big[\mu_n\,\mathrm{accept}\big(\tau_k(\boldsymbol{X}_n), \boldsymbol{X}_n\big)\,\Big|\,\mu_n\Big]$$

Taking expectations with respect to $\mu_n$ on both sides of the above, and recalling definitions (13), (5), (1), (3), (4) proves the claim. Thus, Hoeffding's inequality implies, for all fixed $n, k$,

$$\mathbb{P}\big(\widehat{r}_n^-(k) \leq \mathrm{reward}(\pi_k) \leq \widehat{r}_n^+(k)\big) = \mathbb{P}\Big(\big|\bar{r}_n(k) - \mathrm{reward}(\pi_k)\big| \leq 2\varepsilon_n\Big) \geq 1 - \frac{\delta}{2KN_{\mathrm{ex}}}$$

$$\mathbb{P}\big(\widehat{c}_n^-(k) \leq \mathrm{cost}(\pi_k) \leq \widehat{c}_n^+(k)\big) = \mathbb{P}\Big(\big|\bar{c}_n(k) - \mathrm{cost}(\pi_k)\big| \leq (K-1)\varepsilon_n\Big) \geq 1 - \frac{\delta}{2KN_{\mathrm{ex}}}$$

Applying a union bound shows that event (8) occurs simultaneously for all $n \in \{1, \dots, N_{\mathrm{ex}}\}$ and $k \in \{1, \dots, \max(C_n)\}$ with probability at least $1 - \delta$. $\qquad\square$

**Lemma 6.** *Under the assumptions of Theorem 1, if the event* (12) *occurs simultaneously for all* $n = 1, \dots, N_{\mathrm{ex}}$ *and all* $k = 1, \dots, \max(C_n)$, *and* $\Delta > 0$, *(i.e., if there is a unique optimal policy), then all suboptimal policies are eliminated after at most* $N'_{\mathrm{ex}}$ *tasks, where*

$$N'_{\mathrm{ex}} \leq \frac{288\,K^2\ln(4KN_{\mathrm{ex}}/\delta)}{\Delta^2} + 1 \tag{14}$$

*Proof.* Note first that (12) implies, for all $n \geq 2K^2\ln(4KN_{\mathrm{ex}}/\delta)$ (guaranteed by line 5) and all $k \in C_n$

$$\frac{\widehat{r}_n^-(k)}{\widehat{c}_n^+(k)} \leq \frac{\mathrm{reward}(\pi_k)}{\mathrm{cost}(\pi_k)} \leq \frac{\widehat{r}_n^+(k)}{\widehat{c}_n^-(k)} \qquad \text{if } \widehat{r}_n^+(k) \geq 0$$

$$\frac{\widehat{r}_n^-(k)}{\widehat{c}_n^-(k)} \leq \frac{\mathrm{reward}(\pi_k)}{\mathrm{cost}(\pi_k)} \leq \frac{\widehat{r}_n^+(k)}{\widehat{c}_n^+(k)} \qquad \text{if } \widehat{r}_n^+(k) < 0$$

In other words, the interval

$$\left[\frac{\widehat{r}_n^-(k)}{\widehat{c}_n^+(k)}\mathbb{I}\{\widehat{r}_n^+(k) \geq 0\} + \frac{\widehat{r}_n^-(k)}{\widehat{c}_n^-(k)}\mathbb{I}\{\widehat{r}_n^+(k) < 0\}, \ \frac{\widehat{r}_n^+(k)}{\widehat{c}_n^-(k)}\mathbb{I}\{\widehat{r}_n^+(k) \geq 0\} + \frac{\widehat{r}_n^+(k)}{\widehat{c}_n^+(k)}\mathbb{I}\{\widehat{r}_n^+(k) < 0\}\right]$$

is a confidence interval for the value $\mathrm{reward}(\pi_k)/\mathrm{cost}(\pi_k)$ that measures the performance of $\pi_k$. Let, for all $n, k$,

$$\varepsilon_n = \sqrt{\frac{\ln(4KN_{\mathrm{ex}}/\delta)}{2n}}, \qquad \bar{r}_n(k) = \widehat{r}_n^+(k) - 2\varepsilon_n, \qquad \bar{c}_n(k) = \widehat{c}_n^+(k) - (k-1)\varepsilon_n \tag{15}$$

If $\widehat{r}_n^+(k) \geq 0$, by the definitions in (15), the length of this confidence interval is

$$\frac{\bar{r}_n(k) + 2\varepsilon_n}{\bar{c}_n(k) - (k-1)\varepsilon_n} - \frac{\bar{r}_n(k) - 2\varepsilon_n}{\bar{c}_n(k) + (k-1)\varepsilon_n} = \frac{2\varepsilon_n\big(2\bar{c}_n(k) + (k-1)\bar{r}_n(k)\big)}{\bar{c}_n(k)^2 - (k-1)^2\,\varepsilon_n^2} \leq 12\,K\varepsilon_n$$

where for the numerator we used the fact that $\bar{c}_n(k)$ (resp., $\bar{r}_n(k)$) is an average of random variables all upper bounded by $k$ (resp., 1) and the denominator is lower bounded by $1/2$ because $\bar{c}_n(k)^2 \geq 1$, $(k^2 - 1)\varepsilon_n^2 \leq 1/2$ by $n \geq 2K^2\ln(4KN_{\mathrm{ex}}/\delta)$ (line 4), and $k/K \leq 1$ (by monotonicity of $k \mapsto k$). Similarly, if $\widehat{r}_n^+(k) < 0$, the length of the confidence interval is

$$\frac{\bar{r}_n(k) + 2\varepsilon_n}{\bar{c}_n(k) + (k-1)\varepsilon_n} - \frac{\bar{r}_n(k) - 2\varepsilon_n}{\bar{c}_n(k) - (k-1)\varepsilon_n} = \frac{2\varepsilon_n\big(2\bar{c}_n(k) - (k-1)\bar{r}_n(k)\big)}{\bar{c}_n(k)^2 - (k-1)^2\,\varepsilon_n^2} \leq 12\,K\varepsilon_n$$

where, in addition to the considerations above, we used $0 < -\widehat{r}_n^+(k) < -\overline{r}_n(k) \leq 1$. Hence, as soon as the upper bound $12\,K\varepsilon_n$ on the length of each of the confidence interval above falls below $\Delta/2$, all such intervals are guaranteed to be disjoint and by definition of $C_n$ (line 5), all suboptimal policies are guaranteed to have left $C_{n+1}$. In formulas, this happens at the latest during task $n$, where $n \geq 2K^2\ln(4KN_{\text{ex}}/\delta)$ satisfies

$$12\,K\varepsilon_n < \frac{\Delta}{2} \iff n > 288\,(K/\Delta)^2\ln(4KN_{\text{ex}}/\delta)$$

This proves the result. $\qquad\square$

**Lemma 7.** *Under the assumptions of Theorem 1, if the event* (12) *occurs simultaneously for all* $n = 1, \ldots, N_{\text{ex}}$ *and all* $k = 1, \ldots, \max(C_n)$, *and the test at line 6 is true for some* $N_{\text{ex}}' \leq N_{\text{ex}}$, *then*

$$R_N \leq \min\left(\frac{(2K+1)N_{\text{ex}}}{N}, \ \frac{(2K+1)\big(288\,(K/\Delta)^2\ln(4KN_{\text{ex}}/\delta)+1\big)}{N}\right) \qquad (16)$$

*Proof.* Note that if the test at line 6 is true, than by (12) there exists a unique optimal policy, i.e., we have $\Delta > 0$. We can therefore apply Lemma 6, obtaining a deterministic upper bound $N_{\text{ex}}''$ on the number $N_{\text{ex}}'$ of tasks needed to identify the optimal policy, where

$$N_{\text{ex}}'' = \min\left(N_{\text{ex}}, \ \frac{128\,K^2\ln(4KN_{\text{ex}}/\delta)}{\Delta^2} + 1\right)$$

The total expected reward of Algorithm 1 divided by its total expected cost is lower bounded by

$$\xi = \frac{\mathbb{E}\left[-N_{\text{ex}}' + \sum_{n=N_{\text{ex}}'+1}^{N} \text{reward}(\pi_{k^\star}, \mu_n)\right]}{\mathbb{E}\left[2\sum_{m=1}^{N_{\text{ex}}'} \max(C_m) + \sum_{n=N_{\text{ex}}'+1}^{N} \text{cost}(\pi_{k^\star}, \mu_n)\right]}$$

If $\xi < 0$, we can further lower bound it by

$$\frac{(N - N_{\text{ex}}'')\,\text{reward}(\pi_{k^\star}) - N_{\text{ex}}''}{(N - N_{\text{ex}}'')\,\text{cost}(\pi_{k^\star}) + 2N_{\text{ex}}''} \geq \frac{\text{reward}(\pi_{k^\star})}{\text{cost}(\pi_{k^\star})} - \frac{3N_{\text{ex}}''}{N}$$

where the inequality follows by $(a-b)/(c+d) \geq a/c - (d+b)/(c+d)$ for all $a, b, c, d \in \mathbb{R}$ with $0 \neq c > -d$ and $a/c \leq 1$, and then using $c + d \geq N$ which holds because $\text{cost}(\pi_{k^\star}) \geq 1$. Similarly, if $\xi \geq 0$, we can further lower bound it by

$$\frac{(N - N_{\text{ex}}'')\,\text{reward}(\pi_{k^\star}) - N_{\text{ex}}''}{(N - N_{\text{ex}}'')\,\text{cost}(\pi_{k^\star}) + 2KN_{\text{ex}}''} \geq \frac{\text{reward}(\pi_{k^\star})}{\text{cost}(\pi_{k^\star})} - \frac{(2K+1)N_{\text{ex}}''}{N}$$

Thus, the result follows by $K \geq 1$ and the definition of $N_{\text{ex}}''$. $\qquad\square$

**Lemma 8.** *Under the assumptions of Theorem 1, if the event* (12) *occurs simultaneously for all* $n = 1, \ldots, N_{\text{ex}}$ *and all* $k = 1, \ldots, \max(C_n)$, *and the test at line 6 is false for all tasks* $n \leq N_{\text{ex}}$ *(i.e., if line 7 is executed with* $C_{N_{\text{ex}}+1}$ *containing two or more policies), then*

$$R_T \leq (K+1)\sqrt{\frac{8\ln(4KN_{\text{ex}}/\delta)}{N_{\text{ex}}}} + \frac{(2K+1)N_{\text{ex}}}{N}$$

*Proof.* Note first that by (12) and the definition of $C_n$ (line 5), all optimal policies belong to $C_{N_{\text{ex}}+1}$. Let, for all $n, k$,

$$\varepsilon_n = \sqrt{\frac{\ln(4KN_{\text{ex}}/\delta)}{2n}}, \qquad \overline{r}_n(k) = \widehat{r}_n^+(k) - 2\varepsilon_n, \qquad \overline{c}_n(k) = \widehat{c}_n^+(k) - (k-1)\varepsilon_n \qquad (17)$$

By (12) and the definitions of $k'$, $\widehat{r}_n^{\pm}(k)$, and $\varepsilon_n$ (line 7, (5), (5), and (17) respectively), for all optimal policies $\pi_{k^\star}$, if $\widehat{r}_{N_{\mathrm{ex}}}^{+}(k^\star) \geq 0$, then also $\widehat{r}_{N_{\mathrm{ex}}}^{+}(k') \geq 0$[7] and

$$\frac{\mathrm{reward}(\pi_{k^\star})}{\mathrm{cost}(\pi_{k^\star})} \leq \frac{\widehat{r}_{N_{\mathrm{ex}}}^{+}(k^\star)}{\widehat{c}_{N_{\mathrm{ex}}}^{-}(k^\star)} \leq \frac{\widehat{r}_{N_{\mathrm{ex}}}^{+}(k')}{\widehat{c}_{N_{\mathrm{ex}}}^{-}(k')} \leq \frac{\mathrm{reward}(\pi_{k'}) + 4\varepsilon_n}{\mathrm{cost}(\pi_{k'}) - 2(k'-1)\varepsilon_n}$$

$$\leq \frac{\mathrm{reward}(\pi_{k'})}{\mathrm{cost}(\pi_{k'})} + \frac{2(k'+1)\varepsilon_n}{\mathrm{cost}(\pi_{k'}) - 2(k'-1)\varepsilon_n}$$

where all the denominators are positive because $N_{\mathrm{ex}} \geq 8(K-1)^2 \ln(4KN_{\mathrm{ex}}/\delta)$ and the last inequality follows by $(a+b)/(c-d) \leq a/c + (d+b)/(c-d)$ for all $a \leq 1$, $b \in \mathbb{R}$, $c \geq 1$, and $d < c$; next, if $\widehat{r}_{N_{\mathrm{ex}}}^{+}(k^\star) < 0$ but $\widehat{r}_{N_{\mathrm{ex}}}^{+}(k') \geq 0$ the exact same chain of inequalities hold; finally, if both $\widehat{r}_{N_{\mathrm{ex}}}^{+}(k^\star) < 0$ and $\widehat{r}_{N_{\mathrm{ex}}}^{+}(k') < 0$, then $\widehat{r}_{N_{\mathrm{ex}}}^{+}(k) < 0$ for all $k \in C_{N_{\mathrm{ex}}+1}$[8], hence, by definition of $k'$ and the same arguments used above

$$\frac{\mathrm{reward}(\pi_{k^\star})}{\mathrm{cost}(\pi_{k^\star})} \leq \frac{\widehat{r}_{N_{\mathrm{ex}}}^{+}(k^\star)}{\widehat{c}_{N_{\mathrm{ex}}}^{+}(k^\star)} \leq \frac{\widehat{r}_{N_{\mathrm{ex}}}^{+}(k')}{\widehat{c}_{N_{\mathrm{ex}}}^{+}(k')} \leq \frac{\mathrm{reward}(\pi_{k'}) + 4\varepsilon_n}{\mathrm{cost}(\pi_{k'}) + 2(k'-1)\varepsilon_n}$$

$$\leq \frac{\mathrm{reward}(\pi_{k'})}{\mathrm{cost}(\pi_{k'})} + \frac{2(k'+1)\varepsilon_n}{\mathrm{cost}(\pi_{k'}) + 2(k'-1)\varepsilon_n} \leq \frac{\mathrm{reward}(\pi_{k'})}{\mathrm{cost}(\pi_{k'})} + \frac{2(k'+1)\varepsilon_n}{\mathrm{cost}(\pi_{k'}) - 2(k'-1)\varepsilon_n}$$

That is, for all optimal policies $\pi_{k^\star}$, the policy $\pi_{k'}$ run at line 7 satisfies

$$\mathrm{reward}(\pi_{k'}) \geq \mathrm{cost}(\pi_{k'})\left(\frac{\mathrm{reward}(\pi_{k^\star})}{\mathrm{cost}(\pi_{k^\star})} - \frac{2(k'+1)\varepsilon_n}{\mathrm{cost}(\pi_{k'}) - 2(k'-1)\varepsilon_n}\right)$$

$$\geq \mathrm{cost}(\pi_{k'})\left(\frac{\mathrm{reward}(\pi_{k^\star})}{\mathrm{cost}(\pi_{k^\star})} - 4(K+1)\varepsilon_n\right)$$

where in the last inequality we lower bounded the denominator by $1/2$ using $\mathrm{cost}(\pi_{k'}) \geq 1$ and $\varepsilon_n \leq \varepsilon_{N_{\mathrm{ex}}} \leq 1/2$ which follows by $n \geq N_{\mathrm{ex}} \geq 8K^2 \ln(4KN_{\mathrm{ex}}/\delta)$ and the monotonicity of $k \mapsto k$. Therefore, for all optimal policies $\pi_{k^\star}$, the total expected reward of Algorithm 1 divided by its total expected cost (i.e., the negative addend in (2)) is at least

$$\frac{\mathbb{E}\left[-N_{\mathrm{ex}} + (N - N_{\mathrm{ex}})\,\mathrm{reward}(\pi_{k'})\right]}{\mathbb{E}\left[2\sum_{n=1}^{N_{\mathrm{ex}}} \max(C_n) + (N - N_{\mathrm{ex}})\,\mathrm{cost}(\pi_{k'})\right]}$$

$$\geq \frac{-N_{\mathrm{ex}}}{2\sum_{n=1}^{N_{\mathrm{ex}}} \mathbb{E}\left[\max(C_n)\right] + (N - N_{\mathrm{ex}})\,\mathbb{E}\left[\mathrm{cost}(\pi_{k'})\right]}$$

$$+ \frac{(N - N_{\mathrm{ex}})\,\mathbb{E}\left[\mathrm{cost}(\pi_{k'})\right]}{2\sum_{n=1}^{N_{\mathrm{ex}}} \mathbb{E}\left[\max(C_n)\right] + (N - N_{\mathrm{ex}})\,\mathbb{E}\left[\mathrm{cost}(\pi_{k'})\right]} \left(\frac{\mathrm{reward}(\pi_{k^\star})}{\mathrm{cost}(\pi_{k^\star})} - 4(K+1)\varepsilon_n\right)$$

$$\geq \frac{\mathrm{reward}(\pi_{k^\star})}{\mathrm{cost}(\pi_{k^\star})} - 4(K+1)\varepsilon_n - \frac{N_{\mathrm{ex}} + 2\sum_{n=1}^{N_{\mathrm{ex}}} \mathbb{E}\left[\max(C_n)\right]}{2\sum_{n=1}^{N_{\mathrm{ex}}} \mathbb{E}\left[\max(C_n)\right] + (N - N_{\mathrm{ex}})\,\mathbb{E}\left[\mathrm{cost}(\pi_{k'})\right]}$$

$$\geq \frac{\mathrm{reward}(\pi_{k^\star})}{\mathrm{cost}(\pi_{k^\star})} - 4(K+1)\varepsilon_n - \frac{(2K+1)N_{\mathrm{ex}}}{N}$$

where we used $\frac{a}{b+a}(x - y) \geq x - y - \frac{b}{b+a}$ for all $a, b, y > 0$ and all $x \leq 1$ to lower bound the third line, then the monotonicity of $k \mapsto k$ and $2\mathbb{E}\left[\max(C_n)\right] \geq \mathbb{E}\left[\mathrm{cost}(\pi_{k'})\right] \geq 1$ for the last inequality. Rearranging the terms of the first and last hand side in the previous display, using the monotonicity of $k \mapsto k$, and plugging in the value of $\varepsilon_n$, gives

$$R_T \leq 4(K+1)\varepsilon_n + \frac{(2K+1)N_{\mathrm{ex}}}{N} = (K+1)\sqrt{\frac{8\ln(4KN_{\mathrm{ex}}/\delta)}{N_{\mathrm{ex}}}} + \frac{(2K+1)N_{\mathrm{ex}}}{N}$$

$\square$

---

[7]Indeed, $k' \in \mathrm{argmax}_{k \in C_{N_{\mathrm{ex}}+1}}\left(\widehat{r}_{N_{\mathrm{ex}}}^{+}(k)/\widehat{c}_{N_{\mathrm{ex}}}^{-}(k)\right)$ in this case, and $\widehat{r}_{N_{\mathrm{ex}}}^{+}(k') \geq 0$ follows by the two inequalities $\widehat{r}_{N_{\mathrm{ex}}}^{+}(k')/\widehat{c}_{N_{\mathrm{ex}}}^{-}(k') \geq \widehat{r}_{N_{\mathrm{ex}}}^{+}(k^\star)/\widehat{c}_{N_{\mathrm{ex}}}^{-}(k^\star) \geq 0$.

[8]Otherwise $k'$ would belong to the set $\mathrm{argmax}_{k \in C_{N_{\mathrm{ex}}+1}}\left(\widehat{r}_{N_{\mathrm{ex}}}^{+}(k)/\widehat{c}_{N_{\mathrm{ex}}}^{-}(k)\right)$ which in turn would be included in the set $\left\{k \in C_{N_{\mathrm{ex}}+1} : \widehat{r}_{N_{\mathrm{ex}}}^{+}(k) \geq 0\right\}$ and this would contradict the fact that $\widehat{r}_{N_{\mathrm{ex}}}^{+}(k') < 0$.

## C   A Technical Lemma for Theorem 4

In this section, we give a formal proof for a result needed to prove Theorem 4.

**Lemma 2.** *Let $\Pi$ be a countable set of policies. If ESC is run with $\delta \in (0,1)$, $\varepsilon_1, \varepsilon_2, \ldots \in (0,1]$, and halts returning $K$, then $k^\star \leq K$ for all optimal policies $\pi_{k^\star}$ with probability at least $1 - \delta$.*

*Proof.* Note fist that $\widehat{r}_{2^j}^- + 2\varepsilon_j$ (line 2) is an empirical average of $m_j$ i.i.d. unbiased estimators of $\mathrm{reward}(\pi_{2^j})$. Indeed, being $\mathrm{accept}(k, \boldsymbol{x})$ independent of the variables $(x_{k+1}, x_{k+2}, \ldots)$ by definition of duration and the conditional independence of the samples (recall the properties of samples in step 4 of our online protocol, Section 3), for all tasks $n$ performed at line 2 during iteration $j$ and all $i > 2^j$,

$$\mathbb{E}\left[X_{n,i}\,\mathrm{accept}\left(\tau_{2^j}(\boldsymbol{X}_n), \boldsymbol{X}_n\right)\,\Big|\,\mu_n\right] = \mathbb{E}\left[X_{n,i}\mid\mu_n\right]\mathbb{E}\left[\mathrm{accept}\left(\tau_{2^j}(\boldsymbol{X}_n), \boldsymbol{X}_n\right)\,\Big|\,\mu_n\right]$$
$$= \mu_n\,\mathbb{E}\left[\mathrm{accept}\left(\tau_{2^j}(\boldsymbol{X}_n), \boldsymbol{X}_n\right)\,\Big|\,\mu_n\right] = \mathbb{E}\left[\mu_n\,\mathrm{accept}\left(\tau_{2^j}(\boldsymbol{X}_n), \boldsymbol{X}_n\right)\,\Big|\,\mu_n\right]$$

Taking expectations to both sides proves the claim. Thus, Hoeffding's inequality implies

$$\mathbb{P}\left(\widehat{r}_{2^j}^- > \mathrm{reward}(\pi_{2^j})\right) = \mathbb{P}\left(\left(\widehat{r}_{2^j}^- + 2\varepsilon_j\right) - \mathrm{reward}(\pi_{2^j}) > 2\varepsilon_j\right) \leq \frac{\delta}{j(j+1)}$$

for all $j \leq j_0$. Similarly, for all $l > j_0$, $\mathbb{P}\left(\overline{c}_{2^l} - \mathrm{cost}(\pi_{2^l}) > 2^l\,\varepsilon_l\right) \leq \frac{\delta}{l(l+1)}$. Hence, the event

$$\left\{\widehat{r}_{2^j}^- \leq \mathrm{reward}(\pi_{2^j})\right\} \,\wedge\, \left\{\overline{c}_{2^l} \leq \mathrm{cost}(\pi_{2^l})) + 2^l\,\varepsilon_l\right\} \qquad \forall j \leq j_0, \forall l > j_0 \tag{18}$$

occurs with probability at least

$$1 - \sum_{j=1}^{j_0} \frac{\delta}{j(j+1)} - \sum_{l=j_0+1}^{j_1} \frac{\delta}{l(l+1)} \geq 1 - \delta \sum_{j\in\mathbb{N}} \frac{1}{j(j+1)} = 1 - \delta$$

Note now that for each policy $\pi_k$ with $\mathrm{reward}(\pi_k) \geq 0$ and each optimal policy $\pi_{k^\star}$,

$$\frac{\mathrm{reward}(\pi_k)}{k} \leq \frac{\mathrm{reward}(\pi_k)}{\mathrm{cost}(\pi_k)} \leq \frac{\mathrm{reward}(\pi_{k^\star})}{\mathrm{cost}(\pi_{k^\star})} \leq \frac{1}{\mathrm{cost}(\pi_{k^\star})} \tag{19}$$

Hence, all optimal policies $\pi_{k^\star}$ satisfy $\mathrm{cost}(\pi_{k^\star}) \leq k/\mathrm{reward}(\pi_k)$ for all policies $\pi_k$ such that $\mathrm{reward}(\pi_k) > 0$. Being durations sorted by index, for all $k \leq h$

$$\mathrm{cost}(\pi_k) = \mathbb{E}\left[\mathrm{cost}(\pi_k, \mu_n)\right] \leq \mathbb{E}\left[\mathrm{cost}(\pi_h, \mu_n)\right] = \mathrm{cost}(\pi_h) \tag{20}$$

Thus, with probability at least $1 - \delta$, for all $k > K$

$$\mathrm{cost}(\pi_k) \overset{(20)}{\geq} \mathrm{cost}(\pi_K) \overset{(18)}{\geq} \overline{c}_K - K\,\varepsilon_{\log_2 K} \overset{\text{line 6}}{>} \frac{k_0}{\widehat{r}_{k_0}^-} \geq \frac{k_0}{\mathrm{reward}(k_0)}$$

where $\mathrm{reward}(k_0) \geq \widehat{r}_{k_0}^- > 0$ by (18) and line (3); i.e., $\pi_k$ do not satisfy (19). Therefore, with probability at least $1 - \delta$, all optimal policies $\pi_{k^\star}$ satisfy $k^\star \leq K$.  $\square$

## D   Choice of Performance Measure

In this section, we discuss our choice of measuring the performance of policies $\pi$ with

$$\frac{\sum_{n=1}^N \mathbb{E}\left[\mathrm{reward}(\pi, \mu_n)\right]}{\sum_{m=1}^N \mathbb{E}\left[\mathrm{cost}(\pi, \mu_m)\right]} = \frac{\mathrm{reward}(\pi)}{\mathrm{cost}(\pi)}$$

We compare several different benchmarks and investigate the differences if the agent had a budget of samples and a variable number of tasks, rather than the other way around. We will show that all "natural" choices essentially go in the same direction, except for one (perhaps the most natural) which turns out to be the worst.

At a high level, an agent constrained by a budget would like to maximize its ROI. This can be done in several different ways. If the constraint is on the number $N$ of tasks, then the agent could aim at maximizing (over $\pi = (\tau, \mathrm{accept}) \in \Pi$) the objective $g_1(\pi, N)$ defined by

$$g_1(\pi, N) = \mathbb{E}\left[\frac{\sum_{n=1}^{N} \mathrm{reward}(\pi, \mu_n)}{\sum_{m=1}^{N} \mathrm{cost}(\pi, \mu_m)}\right]$$

This is equivalent to the maximization of the ratio

$$\frac{\mathrm{reward}(\pi)}{\mathrm{cost}(\pi)} = \frac{\mathbb{E}\big[\mathrm{reward}(\pi, \mu_n)\big]}{\mathbb{E}\big[\mathrm{cost}(\pi, \mu_n)\big]}$$

in the sense that, multiplying both the numerator and the denominator in $g_1(\pi, N)$ by $1/N$ and applying Hoeffding's inequality, we get $g_1(\pi, N) = \Theta\big(\mathrm{reward}(\pi)/\mathrm{cost}(\pi)\big)$. Furthermore, by the law of large numbers and Lebesgue's dominated convergence theorem, $g_1(\pi, N) \to \mathrm{reward}(\pi)/\mathrm{cost}(\pi)$ when $N \to \infty$ for any $\pi \in \Pi$.

Assume now that the constraint is on the total number of samples instead. We say that the agent has a *budget of samples* $T$ if as soon as the total number of samples reaches $T$ during task $N$ (which is now a random variable), the agent has to interrupt the run of the current policy, reject the current value $\mu_N$, and end the process. Formally, the random variable $N$ that counts the total number of tasks performed by repeatedly running a policy $\pi = (\tau, \mathrm{accept})$ is defined by

$$N = \min\left\{ m \in \mathbb{N} \,\bigg|\, \sum_{n=1}^{m} \tau(\boldsymbol{X}_n) \geq T \right\}$$

In this case, the agent could aim at maximizing the objective

$$g_2(\pi, T) = \mathbb{E}\left[\frac{\sum_{n=1}^{N-1} \mathrm{reward}(\pi, \mu_n)}{T}\right]$$

where the sum is 0 if $N = 1$ and it stops at $N-1$ because the the last task is interrupted and no reward is gained. As before, assume that $\tau \leq D$, for some $D \in \mathbb{N}$. Note first that by the independence of $\mu_n$ and $\boldsymbol{X}_n$ from past tasks, for all deterministic functions $f$ and all $n \in \mathbb{N}$, the two random variables $f(\mu_n, \boldsymbol{X}_n)$ and $\mathbb{I}\{N \geq n\}$ are independent, because $\mathbb{I}\{N \geq n\} = \mathbb{I}\{\sum_{i=1}^{n-1} \tau(\boldsymbol{X}_i) < T\}$ depends only on the random variables $\tau(\boldsymbol{X}_1), \ldots, \tau(\boldsymbol{X}_{n-1})$. Hence

$$\mathbb{E}\Big[\mathrm{reward}(\pi, \mu_n)\, \mathbb{I}\{N \geq n\}\Big] = \mathrm{reward}(\pi)\, \mathbb{P}(N \geq n)$$

$$\mathbb{E}\big[\mathrm{cost}(\pi, \mu_n)\, \mathbb{I}\{N \geq n\}\big] = \mathrm{cost}(\pi)\, \mathbb{P}(N \geq n)$$

Moreover, note that during each task at least one sample is drawn, hence $N \leq T$ and

$$\sum_{n=1}^{\infty} \mathbb{E}\Big[\big|\mathrm{reward}(\pi, \mu_n)\big|\, \mathbb{I}\{N \geq n\}\Big] \leq \sum_{n=1}^{T} \mathbb{E}\Big[\big|\mathrm{reward}(\pi, \mu_n)\big|\Big] \leq T < \infty$$

$$\sum_{n=1}^{\infty} \mathbb{E}\big[\mathrm{cost}(\pi, \mu_n)\, \mathbb{I}\{N \geq n\}\big] \leq \sum_{n=1}^{T} \mathbb{E}\big[\mathrm{cost}(\pi, \mu_n)\big] = T\, \mathrm{cost}(\pi) \leq TD < \infty$$

We can therefore apply Wald's identity (Wald, 1944) to deduce

$$\mathbb{E}\left[\sum_{n=1}^{N} \mathrm{reward}(\pi, \mu_n)\right] = \mathbb{E}[N]\, \mathrm{reward}(\pi) \qquad \text{and} \qquad \mathbb{E}\left[\sum_{n=1}^{N} \mathrm{cost}(\pi, \mu_n)\right] = \mathbb{E}[N]\, \mathrm{cost}(\pi)$$

which, together with

$$\mathbb{E}\left[\sum_{n=1}^{N} \mathrm{cost}(\pi, \mu_n)\right] \geq T \geq \mathbb{E}\left[\sum_{n=1}^{N} \mathrm{cost}(\pi, \mu_n)\right] - D$$

and

$$\mathbb{E}\left[\sum_{n=1}^{N} \mathrm{reward}(\pi, \mu_n)\right] - 1 \leq \mathbb{E}\left[\sum_{n=1}^{N-1} \mathrm{reward}(\pi, \mu_n)\right] \leq \mathbb{E}\left[\sum_{n=1}^{N} \mathrm{reward}(\pi, \mu_n)\right] + 1$$

yields

$$\frac{\mathbb{E}[N]\operatorname{reward}(\pi) - 1}{\mathbb{E}[N]\operatorname{cost}(\pi)} \le g_2(\pi, T) \le \frac{\mathbb{E}[N]\operatorname{reward}(\pi) + 1}{\mathbb{E}[N]\operatorname{cost}(\pi) - D}$$

if the denominator on the right-hand side is positive, which happens as soon as $T > D^2$ by $ND \ge \sum_{n=1}^N \tau(\boldsymbol{X}_n) \ge T$ and $\operatorname{cost}(\pi) \ge 1$. I.e., $g_2(\pi, T) = \Theta\big(\operatorname{reward}(\pi)/\operatorname{cost}(\pi)\big)$ and noting that $\mathbb{E}[N] \ge T/D \to \infty$ if $T \to \infty$, we have once more that $g_2(\pi, T) \to \operatorname{reward}(\pi)/\operatorname{cost}(\pi)$ when $T \to \infty$ for any $\pi \in \Pi$.

This proves that having a budget of tasks, samples, or using any of the three natural objectives introduced so far is essentially the same.

Before concluding the section, we go back to the original setting and discuss a very natural definition of objective which should be avoided because, albeit easier to maximize, it is not well-suited for this problem. Consider as objective the average payoff of accepted values per amount of time used to make the decision, i.e.,

$$g_3(\pi) = \mathbb{E}\left[\frac{\operatorname{reward}(\pi, \mu_n)}{\operatorname{cost}(\pi, \mu_n)}\right]$$

We give some intuition on the differences between the ratio of expectations and the expectation of the ratio $g_3$ using the concrete example (4) and we make a case for the former being better than the latter.

More precisely, if $N$ decision tasks have to be performed by the agent, consider the natural policy class $\{\pi_k\}_{k \in \{1,\dots,K\}} = \big\{(\tau_k, \operatorname{accept})\big\}_{k \in \{1,\dots,K\}}$ given by

$$\tau_k(\boldsymbol{x}) = \min\left(k, \ \inf\left\{n \in \mathbb{N} : |\bar{x}_n| \ge c\sqrt{\frac{\ln\frac{KN}{\delta}}{n}}\right\}\right), \quad \operatorname{accept}(n, \boldsymbol{x}) = \mathbb{I}\left\{\bar{x}_n \ge c\sqrt{\frac{\ln\frac{KN}{\delta}}{n}}\right\}$$

for some $c > 0$ and $\delta \in (0, 1)$, where $\bar{x}_n = (1/n)\sum_{i=1}^n x_i$ is the average of the first $n$ elements of the sequence $\boldsymbol{x} = (x_1, x_2, \dots)$.

If $K \gg 1$, there are numerous policies in the class with a large cap. For concreteness, consider the last one $(\tau_K, \operatorname{accept})$ and let $k = \lceil c^2 \ln(KN/\delta) \rceil$. If $\mu_n$ is uniformly distributed on $\{-1, 0, 1\}$, then

$$\Big(\tau_K(\boldsymbol{X}_0), \operatorname{accept}\big(\tau_K(\boldsymbol{X}_0), \boldsymbol{X}_0\big)\Big) = \begin{cases} (k, 1) & \text{if } \mu_1 = 1 \\ (k, 0) & \text{if } \mu_1 = -1 \\ (K, 0) & \text{if } \mu_1 = 0 \end{cases}$$

i.e., the agent understands quickly (drawing only $k$ samples) that $\mu_n = \pm 1$, accepting it or rejecting it accordingly, but takes exponentially longer ($K \gg k$ samples) to figure out that the value is nonpositive when $\mu_n = 0$. The fact that for a constant fraction of tasks ($1/3$ of the total) $\pi$ invests a long time ($K$ samples) to earn no reward makes it a very poor choice of policy. This is not reflected in the value of $g_3(\pi_K)$ but it is so in $\operatorname{reward}(\pi_K)/\operatorname{cost}(\pi_K)$. Indeed, in this instance

$$\mathbb{E}\left[\frac{\operatorname{reward}(\pi_K, \mu_n)}{\operatorname{cost}(\pi_K, \mu_n)}\right] = \Theta\left(\frac{1}{k}\right) \qquad \gg \qquad \Theta\left(\frac{1}{K}\right) = \frac{\operatorname{reward}(\pi_K)}{\operatorname{cost}(\pi_K)}$$

This is due to the fact that the expectation of the ratio "ignores" outcomes with null (or very small) rewards, even if a large number of samples is needed to learn them. On the other hand, the ratio of expectations weighs the total number of requested samples and it is highly influenced by it, a property we are interested to capture within our model.

## E  An Impossibility Result

We conclude the paper by showing that, in general, given $\mu_n$ it is impossible to define an unbiased estimator of the reward of all policies using only the samples drawn by the policies themselves, unless $\mu_n$ is known beforehand.

Take a policy $\pi_1 = (1, \operatorname{accept})$ that draws exactly one sample. Note that such a policy is included in all sets of policies $\Pi$ so this is by no means a pathological example. As before, assume for the sake of simplicity that samples take values in $\{-1, 1\}$ and consider any decision function $\operatorname{accept}$ such that $\operatorname{accept}(1, \boldsymbol{x}) = (1 + x_1)/2$ for all $\boldsymbol{x} = (x_1, x_2, \dots)$. In words, the policy $\pi_1$ looks at one single

sample $x_1 \in \{-1, 1\}$ and accepts if and only if $x_1 = 1$. As discussed earlier (Section 2, Repeated A/B testing, and Section D, where $\mu$ is concentrated around $[-1, 0] \cup \{1\}$), there are settings in which this policy is optimal, so this choice of decision function cannot be dismissed as a mathematical pathology.

The following lemma shows that in the simple, yet meaningful case of the policy $\pi_1$ described above, it is impossible to define an unbiased estimator of its expected reward given $\mu_n$

$$\mathbb{E}\big[\mu_n \operatorname{accept}(1, \boldsymbol{X}_n) \mid \mu_n\big] = \mu_n \mathbb{E}\left[\frac{1 + X_{n,1}}{2} \mid \mu_n\right] = \frac{\mu_n + \mu_n^2}{2}$$

using only $X_{n,1}$, unless $\mu_n$ is known beforehand.

**Lemma 9.** *Let $\tilde{X}$ be a $\{-1, 1\}$-valued random variable with $\mathbb{E}[\tilde{X}] = \tilde{\mu}$, for some real number $\tilde{\mu}$. If there exists an unbiased estimator $f(\tilde{X})$ of $\big(\tilde{\mu} + \tilde{\mu}^2\big)/2$, for some $f \colon \{-1, 1\} \to \mathbb{R}$, then $f$ satisfies*

$$\begin{cases} f(-1) = 0 & \text{if } \tilde{\mu} = -1 \\ f(1) = \tilde{\mu} - f(-1)\dfrac{1 - \tilde{\mu}}{1 + \tilde{\mu}} & \text{if } \tilde{\mu} \neq -1 \end{cases}$$

*i.e., to define any such $f$ (thus, any unbiased estimator of $\big(\tilde{\mu} + \tilde{\mu}^2\big)/2$) it is necessary to know $\tilde{\mu}$.*

*Proof.* From $\mathbb{E}[\tilde{X}] = 1 \cdot \mathbb{P}(\tilde{X} = 1) + (-1) \cdot \mathbb{P}(\tilde{X} = -1) = -1 + 2\mathbb{P}(\tilde{X} = 1)$ and our assumption $\mathbb{E}[\tilde{X}] = \tilde{\mu}$, we obtain $\mathbb{P}(\tilde{X} = 1) = (1 + \tilde{\mu})/2$.

Let $f \colon \{-1, 1\} \to \mathbb{R}$ be any function satisfying $\mathbb{E}\big[f(\tilde{X})\big] = \big(\tilde{\mu} + \tilde{\mu}^2\big)/2$. Then, from the law of the unconscious statistician

$$\mathbb{E}\big[f(\tilde{X})\big] = f(1)\mathbb{P}(\tilde{X} = 1) + f(-1)\mathbb{P}(\tilde{X} = -1) = f(1)\frac{1 + \tilde{\mu}}{2} + f(-1)\frac{1 - \tilde{\mu}}{2}$$

and our assumption $\mathbb{E}\big[f(\tilde{X})\big] = \big(\tilde{\mu} + \tilde{\mu}^2\big)/2$, we obtain

$$f(1)(1 + \tilde{\mu}) + f(-1)(1 - \tilde{\mu}) = \tilde{\mu} + \tilde{\mu}^2$$

Thus, if $\tilde{\mu} = -1$, we have $f(-1) = 0$. Otherwise, solving for $f(1)$ gives the result. $\square$