# OpenReview forum: "ROI Maximization in Stochastic Online Decision-Making"
_NeurIPS.cc/2021/Conference — NeurIPS 2021 Poster_

### Official Review · Reviewer_2mFe · 2021-08-05

**Rating:** 7
**Confidence:** 3

**Summary:**

Paper considers a return on investment problem. On round n a task arrives parameterized by a mean parameter mu_n. The learner can observe realizations X_{n} := (X_{n,1},X_{n,2},…) where each is a random variable with mean mu_n as many times as she likes. She uses a policy pi that defines a stopping time tau(X_n) that decides when to stop observing the sequence and then outputs a decision accept(X_n) in {0,1}. The RIO is defined as E[ sum_{n=1}^N mu_n accept(X_n) ] / sum_{n=1}^N E[ tau(X_n) ]. Given a finite set of policies, one can consider the regret as the difference between the optimal policy and the sequence of policies executed by the learner.

A somewhat arbitrary set of policies is considered, with the sole restriction that the kth policy pi_k := (tau_k, accept_k) satisfies tau_k(X_n) <= k almost surely, for all k in a finite set or the natural numbers. The countable set case follows by finding an upper bound K on k*, the index of the optimal policy, and then running their finite set algorithm over K policies.

In the finite setting, if K = O(N^(1/3)) then their regret goes down like K/N^{1/3}. In the countable setting, the regret also scales like this, where now “K” is a random variable returned by their algorithm.

**Main Review:**

Apologies in advance for the brevity of this emergency review. I read the main text in full and also looked at select parts of the appendix (mostly focusing on the impossibility result, detailed explanations of policy sets deferred to the appendix, and verifying the logic of the major steps of the proofs).

Strengths
- Proposed a compelling problem setting
- Connections to related work is comprehensive as far as I am aware
- Clever algorithm to design that combines clearly described concepts
- Generality of policy set

Weaknesses:
- Restrictiveness of policy set?
- Sample complexity seems to have a bad dependence on K. But without a lower bound it is impossible to know.
- Potentially vacuous result in the countable policy class setting.


The algorithm appears primarily motivated by the impossibility result of appendix E, namely, that there exists a policy for which constructing an unbiased estimator of its value by just inspecting its trajectories is impossible. Consequently, the algorithm rolls out each policy at least twice as long as it needs, using the first set of samples to estimate the behavior of the accept rule, and the second set of samples to estimate the nth tasks mean mu_n.

The algorithm exploits the fact that the data from k samples can evaluate policy k (since tau_k <= k), but can also be used to evaluate all policies k’ <= k for free. There are a few algorithmic techniques exploited that are well-described in section 4. The novelty is in the algorithm design which combines these different ideas, rather than the straightforward analysis.

Comments requested from authors:
- The expectation of the ratio of rewards to costs seems more natural than the ratio of their expectations. In the IID setting, presumably these are close, did you consider how things would change if the expectation of the ratio were considered instead?
- The policy set seems both very general (i.e., arbitrary line-crossing stopping times and decision rules) but also restrictive since it is assumed tau_k <= k for all k. Of course, if an infinite set of policies is considered one can always parameterize an arbitrary set of policies by just taking k larger and larger. However, your sample complexity depends on k to have boundedness of the costs, so we cannot actually do this for free. I do not believe Footnote 3, which describes how tau_k <= k can be replaced with tau_k <= D_k fixes this issue since it is not straightforward how the theorems would change if D_k = D_{k’} for many k,k’ and |Pi| >> max_k D_k potentially. The reason why I bring this up is that a class that seems particularly natural would be all stopping times defined by all possible linear boundaries (or monotonically increasing curves), each with their own maximum stopping time D_k. Thus, you may have sub-classes of policies Pi_1,Pi_2,… where the maximum within each sub-class is D_k, but Pi_k may contains many different stopping times. Can you framework be expanded to handle such classes?
- The result of the infinite setting appears somewhat vacuous to me since the returned K is a random variable. And while it is proved that K >= k*, no upper bound on K appears (e.g., in terms of k*) as far as I know, and this K appears in the final theorem. It is very strange to see a theorem that depends not on parameters of the problem, but the behavior of the particular algorithm.

I like this problem statement a lot and enjoyed reading the paper. Given satisfactory clarifications to the above questions, I would be happy to raise my score. But at this point I’m on the fence if these weaknesses are real.

**Time Spent Reviewing:**

4

---

> ### Author Response · Authors · 2021-08-10
> **Response to Reviewer 2mFe**
>
> We sincerely thank the reviewer for agreeing to emergency-review our paper.
> It is deeply appreciated.
>
> ---
>
> - *Expectation of ratio vs. ratio of expectation*
>
> Minimizing the expectation of the ratio
> $$\mathbb E \left[ \frac{\mathrm{reward}(\pi,\mu_n)}{\mathrm{cost}(\pi,\mu_n)} \right]$$
> would push the problem very close to the familiar territory of stochastic bandits.
> However, this modeling choice would not capture a proper ROI-maximization perspective. The reason is that a good policy $\pi$ for ROI maximization should spend as little time as possible on an innovation $\mu_n$ with a small (or null) reward.
> Unlike the ratio of expectations, the expectation of the ratio does not penalize a policy $\pi$ that spends a large number of rounds ($\mathrm{cost}(\pi,\mu_n)$) on a (clearly suboptimal) innovation with $\mathrm{reward}(\pi,\mu_n)\approx 0$ because its contribution gets canceled out by the expectation.
> For more details, see lines 679-699 (end of Section D of the Supplementary Material), where we address this very question.
>
> - *Can $\tau_k \le k$ be replaced by $\tau_k \le D_k$? How would the results change?*
>
> Indeed, it can. There is nothing special in picking $1 \le 2 \le \ldots \le k \le \ldots$ rather than any other increasing sequence of bounds $D_1 \le D_2 \le \ldots \le D_k \le \ldots$.
> By replacing $k$ with $D_k$, Theorems 3 and 4 would scale with $D_K$, rather than $K$.
> The doubt might come from the fact that in our current presentation when the policy set $\Pi$ is finite, $K$ plays the role of both the cardinality of $\Pi$ and a uniform upper bound on their durations. The important bit is the latter, not the former.
> The key intuition on why this is the case is what the reviewer mentioned, that whenever some data is sufficient to evaluate a policy $k$, it can also be used to evaluate all policies $k' \le k$ for free.
> In the example mentioned by the reviewer, where many (even infinitely many) of the durations of the policies share the same upper bound $D_k$, our algorithm would have a very easy time in narrowing down the best one because with only $D_k$ samples, it could keep updated a large number of policy estimators all at once.
> We will clarify this point in the revised version.
>
> - *The role of $K$ in the statement of Theorem 4*
>
> You are correct in saying that the random variable $K$ appearing in Theorem 4 is the outcome of a previous algorithm. We stated the bound that way because $K$ has a natural interpretation as the "effective" number of policies. That said, we take your point that it is somewhat unconventional (and potentially vacuous) to have the bound scale with an algorithm-dependent quantity. Below, we describe how to get rid of this issue with a simple argument.
>
> First, note that if a lower bound $\lambda^*$ on the ROI of an optimal policy were known beforehand, then it would be easy to control $K$  by estimating the cost of policies $2^\ell$, for increasing values of $\ell$ (as in ESC). Hence $K$ would be upper-bounded by the number of policies whose cost is smaller than $2/\lambda^\star$ (say). Note that this quantity is now constant (in $N$, as it only depends on the distribution $\mu$ and the set of policies $\Pi$, playing the role of a complexity measure of the problem instance).
> Then, if $\lambda^\star$ is unknown, one can first estimate it up to some multiplicative constant (say, $1/2$), with high probability. Again, this can be done at a constant cost (depending only on the instance $(\mu,\Pi)$) via a multiplicative Chernoff bound. We will certainly mention this improvement in the revised version, and we thank the reviewer for suggesting it!
>
> ---
>
> We thank again the reviewer for their comments and suggestions. We are happy to answer any other follow-up questions (if any) during the discussion period. If all doubts were resolved, we would be happy if the reviewer could consider raising their score.

---

> > ### Comment · Reviewer_2mFe · 2021-08-25
> > **Response**
> >
> > Thank you for your response -- you've addressed most of my concerns. I still don't understand how you can estimate lambda* efficiently (i.e., in a way that doesn't depend on the policy gap). Nevertheless, I've seen enough to give me confidence that the authors will make such points clearer in the final.

---

### Official Review · Reviewer_secW · 2021-08-06

**Rating:** 5
**Confidence:** 3

**Summary:**

This paper studies the problem of Return On Investment (ROI) maximization in repeated decision-making. The main contributions are as follows.

The authors first propose a framework for ROI maximization in Sec. 3. The main contribution here is to formulate an objective/regret in Eq. (2), which is a ratio of reward over cost. The problem is then to maximize the average reward-cost ratio of $N$ tasks, where each task's value is an i.i.d. sample from a true $\mu$.

When the policy class is finite with $K > 0$ policies, the authors propose the Capped Policy Elimination (CAPE) in Algorithm 1, and show that its convergence rate is $\tilde{O}(1/N^{1/3})$ in Theorem 1. The main idea is to make a decision first, then use oversampling to get i.i.d. samples of the reward and cost, and then use concentration inequality to estimate the performance gap $\Delta$ of policies to do policy elimination.

When the policy class is countably infinite, the authors propose the : Extension to Countable (ESC) in Algorithm 2 to reduce the policy class to finite. The main idea is to estimate the upper bound of the optimal policy's cost, such that any optimal policy would take smaller number of observations. Then CAPE is used on the reduced policy class, and a similar $\tilde{O}(1/N^{1/3})$ convergence rate upper bound is shown in Theorem 4.



**Limitations And Societal Impact:**

The results are theoretical, and I did not see any potential negative societal impact of this work.

**Main Review:**

Disclaimer: My sincere apologies for the late review, since I have been assigned as an emergency reviewer for this paper. This makes it possible that I could misunderstand some details of this work.

Overall, the presentation is clear and easy to follow. The motivation is clear and the intuition of the algorithm makes sense. The proofs are mostly correct as I have checked.

I have the following questions that I would ask the authors to clarify.

1. I am confused with the assumption/setting of "the value $\mu_n$ of the current innovation is drawn i.i.d. according to $\mu$".

Does this mean the $N$ tasks/innovations are actually all copies of a single task with true value $\mu$, and the goal is then to estimate the one true value $\mu$? If this is the case, this seems to be easier than a bandit problem, where the true values of all the actions are independent? Or this can be value as a "weird" specific bandit problem: we have different actions, but their rewards are all noisy copies of one true value. This assumption seems unnatural to my understanding. Please comment on this and what would be the result if this assumption is removed?

2. There seems to be another possibility of view this problem as an bandit problem: each policy is an action, and each action's value is then the policy's reward-cost ratio. Then the problem is to choose the best action (policy) with the highest true value. This seems to make more sense. Please clarify or correct my understanding.

3. If my understanding is correct, then I feel the techniques are standard and the results are not very surprising. The reward is bounded in between $[-1, 1]$, and the cost is integers. This makes the problem a bit different but very similar to just estimating the reward: since the cost is unbounded positive integer, one just need to estimate the reward when the cost is not too large. Otherwise any large cost would rule out the optimal policy since the reward is a bounded value.

Therefore the optimal ratio can be figured out just using estimation of both the numerator and denominator as the authors did. This makes it like an extension of explore-then-commit strategy, using empirical means and concentration inequalities. Although oversampling is used to get i.i.d. estimation, the main idea is basically do mean estimation for both reward and cost.

Given there is no lower bound (the authors conjectured that their upper bound is optimal), it would be more convincing if the authors could study other designs similar to the UCB algorithm in bandit to see if a better upper bound could be obtained, or directly show some information-theoretic lower bound using the related results as noted.

4. The ESC strategy looks interesting, but it seems it highly relies on the fact that the reward is bounded and cost is positive integer (as also noted the optimal policy cannot have too large cost). Is it possible that for other cost functions (which are not necessarily like positive integers here), where there are countably infinite actions with both bounded reward and bounded cost, a similar procedure of reducing the policy class to a finite number can be designed?



**Time Spent Reviewing:**

8

---

> ### Author Response · Authors · 2021-08-10
> **Response to Reviewer secW**
>
> We sincerely thank the reviewer for agreeing to emergency-review our paper.
> It is deeply appreciated.
>
> ---
>
> Before answering each specific question, we quickly recall the setting.
>
> For each task $n$, the environment secretly draws a sample $\mu_n \in [-1,1]$ i.i.d. according to some unknown, fixed distribution.
> This $\mu_n$ represents the hidden value of the innovation that the learner has to evaluate during their $n$-th task.
> $\mu_n$ is never revealed to the learner, but the learner can draw any number of i.i.d. samples $X_{n,1}, X_{n,2},\dots$ (with mean $\mu_n$) before stopping and deciding to either accept or reject the current innovation.
> If the learner accepts, they gain the current value $\mu_n$; otherwise, they get zero. The performance of a sequence of decisions is measured as the (expected) total value accumulated by accepting innovations divided by the (expected) total number of samples drawn. The goal is to get this performance as close as possible to the best ROI$(\pi)$ of a policy $\pi$ belonging to some class.
>
> At a high level, we are not interested in learning precisely the law that governs the values $\mu_n$ of the innovations, nor we can afford a large initial effort to find the very best policy. Rather, we want to accept as many good innovations as possible in the shortest amount of time.
> Note that our solution concept is not to match the best innovation $n$ (as is the case in Multi-Arm Bandit regret minimization). We are not interested in learning the expectation of $\mu_n$, and even if the expectation is negative our learner can have a positive outcome (by accepting innovations that are likely to have positive value, given the experimentation).
>
> This model fits the need of companies (e.g., tech companies) that have active R\&D teams, constantly proposing new ideas for innovation. For these companies, it is not important to understand how all the hidden parameters of the market work, nor do they have the luxury of slowing down the flow of progress to fine-tune an optimal policy offline. Rather, they need to enact a sequential strategy that is demonstrably able to gauge if each innovation is beneficial or not with a minimal amount of testing.
>
>
> 1. In our model, $\mu_1,\mu_n,\ldots$ is a sequence of i.i.d. *random variables*. We denoted their common *distribution* by $\mu$, which perhaps was not the best choice, as it caused some confusion. We will replace $\mu$ with $D$ in the revised version to make it crystal clear that $\mu_n$ and $D$ are different objects. We thank the reviewer for pointing out this possible misunderstanding. For more details on this assumption, we refer the reviewer to lines 46-57. As far as lifting this assumption goes, the next natural line of research is to consider distributions of $\mu_n$ that vary over time (we mention this in lines 101-103). Consider, e.g., an adversarial setting. Given the unique form of the problem, we are not sure yet what types of results we could expect in such a case. In fact, it is not even clear if convergence to optimal ROI is possible when playing in an adversarial environment.
>
> 2. In stochastic bandits, after the learner pulls an arm, an independent realization of its reward is revealed, and the goal is to minimize the cumulative regret. If we think of the policies as "arms" and their "ROI" as the corresponding reward, we first note that pulling an "arm" *does not* reveal an independent realization of its "reward" (in fact, in Section E of the Supplementary Material, we show more: that it is demonstrably impossible to build an unbiased estimator of $\mathrm{ROI}(\pi)$ based only on the data coming from "pulling" $\pi$). Even worse, our objective is much more punishing than cumulative regret. To see it, note that if the rewards in bandits are bounded in $[-1,1]$, pulling a bad arm once costs at most 2, i.e., it is always negligible in terms of regret. In our setting, pulling a bad "arm" (i.e., running a long and suboptimal policy) *just once* could have catastrophic effects, driving the algorithm's performance all the way down to near zero. We refer the reviewer to lines 131-166 for additional details on the differences between these two problems.
>
> 3. Again, we stress that even the easier task of estimating the rewards of our policies requires some novel ideas because of the impossibility result of Appendix E. Furthermore, concentration inequalities are but one of the tools that we use in our proof: there are many moving parts that one has to bring together to get to our results. In particular, a straightforward adaptation of explore-then-commit would not suffice to prove Theorem 1. We refer the reviewer to lines 256-281 for a discussion on why this is the case. Regarding a UCB strategy, we do not have an impossibility result, but all evidence points to it not working (another striking difference with stochastic bandits). UCB works in bandits because pulling an arm reveals an unbiased estimate of its reward. The more one pulls an arm, and the more its corresponding mean concentrates around its expectation. Due to the inherent bias present in our setting, the estimated reward of an optimal policy would concentrate around some arbitrary value with no guarantees that the corresponding *biased* ROI would still be the highest among all *biased* ROIs. The only way out seems to be recovering unbiased estimates, and (because of our impossibility) the only way to recover unbiased estimates is through oversampling, which essentially makes one fall back to the paradigm of our CAPE algorithm.
>
> 4. We precise that only the *expected* reward of a policy needs to be bounded for the observation at lines 326-331 to apply (as can be seen from lines 327-328), which is significantly less than assuming that the reward itself (defined in (1)) is. As we mentioned in footnote 1, one can relax the boundedness of $\mu_n$ to subgaussianity. This relaxation would (in general) yield unbounded rewards, but the theory would still hold. Regarding costs of policies $\pi_k$, we defined them (in (1)) as the (identity applied to the) duration $\tau_k(\boldsymbol X_n)$ of a run. Following our exact proofs, one can see that everything keeps holding if we swap the identity with any increasing function of the duration. This essentially puts no restrictions on cost functions, as non-monotone cost functions would not make much modeling sense. We thank the reviewer for pointing out this improvement. We will make sure to mention it in the revised version.
>
> ---
>
> We thank again the reviewer for their comments and suggestions. We are happy to answer any other follow-up questions (if any) during the discussion period. If all doubts and misunderstandings were resolved, we would be happy if the reviewer could consider raising their score.

---

> > ### Comment · Reviewer_secW · 2021-09-03
> > **There is something still unclear/unconvincing to me**
> >
> > I would like to thank the authors for the feedback. I have read them as well as the other reviews. That helped me a lot get a better understanding of this work.
> >
> > However, I am still not convinced since the following are not clear to me based on the feedback.
> >
> > The main difficulty (and why existing results like bandits cannot be trivially applicable here) of the ROI maximization problem studied in this paper, as summarized, are (i) impossibility results for one cannot get an unbiased estimator as shown in the appendix; (ii) the nature of ROI is a ratio rather than additive/cumulative regret used in bandit literature.
> >
> > For the first point (i) impossibility results, as pointed by another reviewer etbW (as also confirmed by the authors in their feedback), is "relatively easy to oversample and build an estimator", and the authors considered this is right so they "did not include it in the main body for similar reasons".
> >
> > For the second point (ii) ROI is a ratio, how is maximizing a ratio $\frac{ \text{reward}(\pi) }{ \text{cost}(\pi) }$ different with maximizing its logarithm, i.e., $\log{ ( \text{reward}(\pi) ) } - \log{ ( \text{cost}(\pi) ) }  $, which makes the objective additive?
> >
> > I think there is no big difference. For example, in supervised learning, people do not consider maximizing the likelihood strictly more difficult than maximizing the log-likelihood, since there are essentially equivalent. Moreover, if maximizing the multiplicative likelihood itself is more difficult, then that means it is not necessary to do that (a simpler objective shares the same solutions, and people should consider the additive objectives).
> >
> > And the only difficulty I can imagine here is what if the expected reward $\text{reward}(\pi)$ is close to $0$, making the $\log{ ( \text{reward}(\pi) ) }$ negatively unbounded. This could also be resolved by just shifting the reward and assuming $\text{reward}(\pi)$ bounded away from $0$ without changing the difficulty of the problem, e.g., $\text{reward}(\pi) \in [ 1, 2] $.
> >
> > So if the above is true, the two main difficulties seem to be not real difficulties, and they are more artificial to me. Based on the my current understanding, I would choose to keep my current rating.
> >
> > And please correct me if I was still misunderstanding something important.

---

> > > ### Author Response · Authors · 2021-09-04
> > > **Response**
> > >
> > > We thank the reviewer for their questions.
> > >
> > > (i) Regarding the response to the other reviewer, what we meant is that the *proof* of the impossibility result is relatively simple and perhaps not worthy of the main body. Yet, its *consequences* are noteworthy. The lemma gives strong evidence that to get results, some oversampling is required. This hurdle rules out direct applications of bandit techniques because oversampling a policy even by one round lowers its ROI by a multiplicative constant (bounded away from 1), preventing convergence to the optimal ROI (unless one employs some capping procedure, as we suggest doing in our CAPE algorithm).
> > >
> > > (ii) Please note that assuming that $\mathrm{reward}(\pi)$ is positive and bounded away from zero would *drastically* change the difficulty of the problem. More than that, it would trivialize it! If all rewards are positive, the best policy (which we can always assume is in $\Pi$) is simply the one that automatically accepts all innovations after just 1 sample. The fact that the rewards could be zero or negative is a crucial aspect of the problem. Indeed, the whole point of the setting is to determine an efficient way to get rid of non-positive innovations quickly. We hope this clarifies that the inherent complexities that arise in our natural model of sequential decision-making cannot be bypassed with simple reductions to known settings (and, in particular, why the $\log$ argument proposed by the reviewer fails).
> > >
> > > ---
> > >
> > > We thank the reviewer again for engaging in the discussion. We are happy to answer any other follow-up questions. If we conclusively cleared all doubts and misunderstandings, we would be happy if the reviewer could consider raising their final score.

---

### Official Review · Reviewer_etbW · 2021-08-07

**Rating:** 7
**Confidence:** 4

**Summary:**

This paper considers a sequential decision process that models that of ideation. At any time an agent has access to a reward distribution and they can either take a sample from the distribution, accept the distribution and receive a reward equivalent to the mean of the distribution, or reject the distribution and receive the reward of 0. The goal of the agent is to find a policy that maximizes their expected ROI, that is the ratio of the total reward of their policies over the total cost (expected number of samples a policy takes).

**Limitations And Societal Impact:**

Please see above.

**Main Review:**


Overall I thought the paper was interesting and studied an interesting problem statement. I did have a few concerns.

1. Organizational comments: I found the background material in Section 2 a bit difficult to parse since I had not fully understood the precise definition of ROI by the time I read this section. Maybe the author can switch the order of 2 and 3?

2. My main theoretical concern is regarding the guarantees of Escape in Theorem 4. The K given in the statement of the upper bound is not explicit and hence the bound seems somewhat useless as a result. I also struggled to understand the tradeoff between $\epsilon$ and $K$ in ESC. Presumably, it depends on the underlying distribution? Maybe the authors can clarify?

3. Finally, the authors imply that the inability to compute an unbiased estimate of a policy is a significant obstacle to overcome. However, as they show, it’s relatively easy to oversample and build an estimator. This seems more of a remark (albeit an interesting one) than a real obstacle to highlight as a contribution.



**Time Spent Reviewing:**

4

---

> ### Author Response · Authors · 2021-08-10
> **Response to Reviewer etbW**
>
> We sincerely thank the reviewer for agreeing to emergency-review our work and for sharing their comments. It is deeply appreciated.
>
> ---
>
> 1. Thank you for bringing this to our attention. If the space constraints allow it, we will add more details to lines 34-45, such as an explicit formula for the ROI. If it is not possible, we will, as suggested, push Section 2 after Section 3.
>
> 2.
> - We take your (and another's reviewer) point that it is somewhat unconventional to have the bound in Theorem 4 scale with an algorithm-dependent quantity (namely, $K$). Fortunately, a simple argument gets rid of this issue.
> Note that if a lower-bound $\lambda^*$ on the ROI of an optimal policy were known beforehand, then it would be easy to control $K$  by estimating the cost of policies $2^\ell$, for increasing values of $\ell$ (as in ESC). In this case $K$ would be upper-bounded by the number of policies whose cost is smaller than $2/\lambda^\star$ (say). Note that this quantity is now constant (in $N$; it depends only on the distribution $\mu$ and the set of policies $\Pi$, playing the role of a complexity measure of the problem instance).
> If $\lambda^\star$ is unknown, one can first estimate it up to some multiplicative constant (say, $1/2$), with high probability. Again, we can do this at a constant cost (depending only on the instance $(\mu,\Pi)$) via a multiplicative Chernoff bound. We will certainly mention this improvement in the revised version, and we thank the reviewer for suggesting it.
>
> - Regarding the trade-off between $\varepsilon$ and $K$, look first at line 3 of ESC. The **if** clause checks whether a confidence interval (for reward$(\pi_{2^j})$) of length $\approx \varepsilon$ sits to the right of the origin. For this to happen, $\varepsilon$ has to be small (smaller than the expectation we are estimating, reward$(\pi_{2^j})$). If this is not the case, the **if** will always be false (with high probability), and the algorithm will not halt. This observation leads to taking $\varepsilon$ small, but from the initialization, we see that $\varepsilon$ small would result in a long running-time for a different reason (because it directly controls how many tasks we are burning through at line 2, since $m_j \approx 1/\varepsilon^2$). Hence $\varepsilon$ has to be selected in a sweet spot where it is small enough so that the **if** can be true, but it is large enough so that we don't run out of tasks before entering line 4.
> We are happy to answer any follow-up questions in case there are some lingering doubts. We will slightly update the discussion at lines 365-374 in light of this feedback.
>
> 3. We essentially agree with the reviewer. We did not include it in the main body for similar reasons. The key message of the impossibility result (and the reason why it deserves at least some attention) is that it gives a strong indication that to get results, some oversampling is required, with all the hurdles that this realization brings.

---

> > ### Comment · Reviewer_etbW · 2021-09-12
> > **acknowledged**
> >
> > Thanks for the clarifications. I will leave my review as is.

---

### Decision · Program_Chairs · 2021-09-27

**Decision:**

Accept (Poster)

**Comment:**

The paper considers a variant of a sequential selection problem under "partial feedback." This is formulated and presented as a variant of an ROI problem. The main result is an algorithm for implementing the selection rule and its convergence rate analysis. The main difficulty is discerning when to stop the experimentation and move to the next stage. Two of the reviewers are positive about the paper and one is more negative in his/her remarks. Generally there is an appreciation for the problem being studied and its broad relevance and interest. (A good chance to thank all referees for their invaluable input.)  I think this is an interesting paper and has the potential to be relevant for NeurIPS audience. I will go along with the two more positive reviews, although my personal view on the paper is that it still has some major issues and I believe it to be a borderline accept at best. I hope some of these issues  can be addressed in the revision.

1. In my view the general "motivating story" as well as the selected title overpromises and this sets the tone for the paper.  The sequential technology testing story is indeed something many companies engage in, but I seriously doubt whether any reasonable practical process resembles the objectives in the paper where the company has a pre-defined set of policies and it tries to achieve result close to the "best" in this class. This type of logic, similar to the best action in hindsight, and the "expert" paradigm in ML is more of theoretical analysis value. The latter is a meaningful analysis framework, but connecting it through this motivating example is tenuous at best. It leaves a bit of a sense of an answer searching for a problem. While the paper studies online decision making, certainly the key results only pertain to a rather restricted canvass in that space. For example,  multistage  A/B testing may better reflect what the paper is modeling and analyzing.  I would hope the authors can address this in their revision.

2. If the idea is to connect to such practical settings, why not take a simple class of parametrized policies and analyze that? The problem can be fairly  easily formulated as a sequential selection problem for which there is rich literature and plenty of heuristics to choose from. For example, in such settings it is common, as in classical PAC learning, to select a tolerance parameter, say delta, and study the run length of sequential experiments needed to conclude with high probability  that the true mean is positive (or exceeds, some epsilon, say). It is then possible to study this as a heuristic with epsilon set to shrink with sampling horizon. This would be somewhat akin to a modified best arm identification problem, but with the added element of having multiple stages in the game (see further comments below).  The current formulation seems to me a bit contrived, especially that it effectively requires a finite cardinality policy class. (The extensions discussed in the paper notwithstanding.)

3. I also feel the placement of the contribution relative to literature is sketchy in parts. The problem is in essence a sequential selection problem. Perhaps i'm missing something, but I don't see a direct connection to the prophet inequality literature. That literature  either seeks to approximate state-dependent stopping rules with single (or multiple) threshold rules, or attempts to bound losses of expected optimal performance relative to the best offline solution in a horizon independent manner. If this is supposed to represent the sequential selection literature it is not the best connective fabric. There is vast literature on optimal stopping problems that have multiple stages that are far more relevant for example. The connection to the bandit problem is also superficial. Each stage is more connected to a best arm identification in an environment where there is a known alternative that has mean zero. As such the complexity analysis in Emilie  Kauffman's work, in particular the recent paper with Garvier on sequential hypothesis testing with overlapping hypotheses seems more relevant.   Rather than screening less relevant strands I would suggest to focus more on the core areas that intersect with this problem and then clearly explain how the paper adds to existing results. (As an additional example, the recent papers on A/B testing by Schmit et al and Azevedo et al are mentioned in passing only...)

4.  The absence of a lower bound argument leaves the analysis incomplete (the authors' conjecture aside). But I recognize this is beyond the scope of the revision.

Finally, in conjunction with 1.) I believe assigning  a more modest (and descriptive)   title and packaging aligned with contribution would also be a welcome modification in the revision.